# A Language-Guided Bayesian Optimization
# for Efficient LoRA Hyperparameter Search

Baek Seong-Eun [1]   Lee Jung-Mok [2]   Kim Sung-Bin [3]   Tae-Hyun Oh [4]

## Abstract

Fine-tuning Large Language Models (LLMs) with Low-Rank Adaptation (LoRA) offers a resource-efficient way to personalize or specialize. However, LoRA is highly sensitive to hyperparameter choices, and exhaustive hyperparameter search is computationally expensive. To address this, we propose a Bayesian Optimization (BO) framework that leverages the domain knowledge of pretrained LLMs to efficiently search for LoRA hyperparameters. Our approach repurposes a pretrained LLM as a discrete-to-continuous mapping module to link hyperparameters and their domain knowledge to a continuous vector space, where BO is conducted. We design and control the mapping via language prompting, providing a domain-aware textual prompt that describes the relationships among hyperparameters and their respective roles. This allows us to explicitly inject domain knowledge about LoRA into the LLM in natural language. We also introduce an additional learnable token to capture residual information that is difficult to describe linguistically in the prompt. This aids BO to sample more high-performing hyperparameters. In addition, by leveraging the strong correlation observed between the performance obtained from full and subset training datasets in LoRA training regimes, we introduce proxy training and evaluation using a data subset. This significantly improves the efficiency of our method. We demonstrate that our hyperparameter, discovered with only about 30 iterations, achieves more than 20% performance improvement over standard hyperparameters found from about 45,000 combinations. Project page: https://baekseongeun.github.io/lora-bo/

[1]Grad. School of AI, POSTECH, Pohang, Korea [2]School of EE, KAIST, Daejeon, Korea [3]Dept. of EE, POSTECH, Pohang, Korea [4]School of Computing, KAIST, Daejeon, Korea. Correspondence to: Tae-Hyun Oh <taehyun.oh@kaist.ac.kr>.

*Proceedings of the 43rd International Conference on Machine Learning*, Seoul, South Korea. PMLR 306, 2026. Copyright 2026 by the author(s).

## 1. Introduction

Large Language Models (LLMs) (Touvron et al., 2023; Team, 2024; Team et al., 2024) have been recognized as strong foundation models that can be easily adapted to diverse downstream tasks with high performance. However, fully fine-tuning LLMs for specific applications is computationally heavy. It requires updating billions of parameters, which demands substantial memory and computational resources (Brown et al., 2020; Gururangan et al., 2020). To overcome these limitations, Parameter-Efficient Fine-Tuning (PEFT) (Houlsby et al., 2019; Hyeon-Woo et al., 2022; Ding et al., 2023) methods have emerged as effective alternatives, enabling strong task adaptation at significantly reduced cost. Among these approaches, Low-Rank Adaptation (LoRA) (Hu et al., 2022) stands out as one of the most widely adopted techniques. LoRA freezes the pretrained weights and introduces lightweight, trainable low-rank adapters, allowing models to adapt efficiently to new tasks with only a fraction of the parameters and resources required for full fine-tuning.

Despite its effectiveness, identifying an *near-optimal* hyperparameter for LoRA remains challenging, as performance is highly sensitive to hyperparameter choices (Sengupta et al., 2024; Biderman et al., 2024; Mao et al., 2025). LoRA involves several key hyperparameters, including the rank ($r$), scaling factor ($\alpha$), batch size, learning rate, and dropout rate, which are entangled in complex ways. Consequently, performance can vary significantly depending on their combinations (Halfon et al., 2024; Sengupta et al., 2024; Mulakala et al., 2024). LoRA is typically applied to LLMs and exhibits optimization characteristics that differ from those of conventional machine learning models, making generic hyperparameter optimization (HPO) strategies less directly applicable and motivating LoRA-specific treatment. Therefore, systematically searching for the appropriate configuration is critical. However, naïve exploration is infeasible: the hyperparameter search space is too large, and each evaluation is extremely costly (Valipour et al., 2022; Chavan et al., 2023; Sun et al., 2024; Meo et al., 2024; Bini et al., 2025).

This challenge motivates the use of Bayesian optimization (BO) as a principled framework for HPO. BO has proven highly effective in real-world applications where target

function evaluations are expensive, such as drug discovery and materials design (Korovina et al., 2020; Ranković & Schwaller, 2023; 2025). BO relies on a surrogate model to approximate the black-box function defined by hyperparameters and their performance, and uses an acquisition function to select the next configuration by balancing exploration and exploitation. However, directly applying BO to LoRA HPO is non-trivial, since traditional BO methods struggle to incorporate domain knowledge into the optimization process (Yan et al., 2025) and require the underlying function domain to be continuous and smooth while the hyperparameter space of our interest is discrete. Additionally, BO relies on surrogate models trained with limited data, which can lead to ineffective optimization if the surrogate model is poorly learned.

In this work, we propose an efficient BO-based HPO framework tailored to LoRA. This framework leverages the prior knowledge in LLM to enhance the optimization process and takes advantage of LLM's ability to encode textual information, enabling domain knowledge integration and providing flexibility in adapting to changes in input. Specifically, hyperparameter configurations are expressed as structured text templates, which describe each hyperparameter's name, value, role, and interactions. An LLM processes this template along with a learnable token and converts it into a continuous embedding, where domain knowledge is effectively encapsulated in the learnable token. The learnable token, combined with observed performance data, is then used to train a surrogate model. This model proposes hyperparameter candidates that maximize the acquisition function. To further improve efficiency, we introduce a proxy training evaluation that reduces evaluation cost and iteration time, enabling faster and more sample-efficient optimization.

Our framework generalizes beyond LoRA to its variants, including DoRA (Liu et al., 2024b), rsLoRA (Kalajdzievski, 2023), and PiSSA (Meng et al., 2024), and is compatible with various model architectures. Experimental results show consistent performance improvements when applying our HPO framework across diverse settings. Additionally, our approach proves both more efficient and effective than existing search methods (Oliver & Wang, 2024; Tribes et al., 2024) and alternative optimization strategies (Bergstra & Bengio, 2012; Akiba et al., 2019; Li et al., 2021). Finally, by analyzing the results, we observe that previously unexplored hyperparameter combinations can also lead to strong performance, offering new insights into LoRA hyperparameters.

In summary, our contributions are as follows:

- **The first framework combining an LLM with BO specialized for LoRA HPO.** We propose an efficient BO-based LoRA HPO framework that integrates domain knowledge into the optimization process through an LLM, enabling the selection of appropriate hyperparameters from a vast number of possible combinations.

- **Improving efficiency of the proposed framework.** We introduce a projection layer and a learnable token to accelerate the BO process. We also introduce a proxy training evaluation protocol that significantly reduces computational cost, enabling efficient optimization.

- **Empirical validation of efficiency and generalizability.** We demonstrate the generalizability of our framework across LoRA variants and model architectures, showing consistent improvements and offering new insights into effective hyperparameter configurations.

## 2. Related Work

**Low-Rank Adaptation (LoRA) and hyperparameter sensitivity in LoRA.** LoRA (Hu et al., 2022) has become one of the most widely adopted parameter-efficient fine-tuning (PEFT) methods (Houlsby et al., 2019) for Large Language Models (LLMs). By introducing a trainable low-rank adapter into a frozen pre-trained model, LoRA allows efficient task-specific adaptation without updating the full set of model parameters. Building on this idea, various LoRA variants have been proposed to improve stability, convergence, and performance. For example, DoRA (Liu et al., 2024b) decomposed each weight into a fixed magnitude and a learnable low-rank direction and rsLoRA (Kalajdzievski, 2023) rescaled LoRA updates by a factor of $\alpha/\sqrt{r}$ to improve stability. Meng et al. (2024) suggest PiSSA leveraging the principal singular vectors and values of the original weights to initialize LoRA adapters for faster convergence and performance improvement.

Although several advanced LoRA variants have been proposed, the common issue of sensitivity to hyperparameter selection remains a challenge. In particular, rank ($r$) (Zhang et al., 2024), scaling factor ($\alpha$) (Liu et al., 2025), learning rate (Jin et al., 2023), batch size (Marek et al., 2025), and dropout rate (Lin et al., 2024) identified as key factors that influence final results. This sensitivity often leads to large performance variance and complicates fair comparisons across methods. Moreover, the "near-optimal" configuration frequently depends on the dataset and base model in use (Rajabzadeh et al., 2024; Yan et al., 2025). Consequently, systematic approaches for optimizing LoRA hyperparameters remain underexplored, as it is difficult to identify effective configurations while accounting for all these factors. Prior work has explored black-box optimization methods (Inouye et al., 2024; Tribes et al., 2024; Oliver & Wang, 2024; Sengupta et al., 2024) and efficient grid-search strategies (Yan et al., 2025) for LoRA hyperparameter selection. Nevertheless, these approaches commonly suffer from two limitations: (i) domain knowledge is not incorporated into the optimization process, and (ii) evaluation remains costly. Hyperparameter optimization generally requires substantial

domain knowledge (Wu et al., 2019; Shawki et al., 2021; Czako et al., 2021; Bowler et al., 2022), and LoRA is no exception due to its adapter-specific properties (Halfon et al., 2024; Yan et al., 2025). To address these limitations, we propose a framework that integrates Bayesian optimization and an LLM. This framework can automatically and effectively identify suitable hyperparameters for LoRA, reducing the need for extensive manual tuning.

**Bayesian optimization for hyperparameter optimization**. Hyperparameter optimization is a critical task that significantly impacts model performance in machine learning. However, evaluating each configuration is often expensive due to the high cost of training. In this context, Bayesian optimization has emerged as a prominent method for HPO, especially in expensive evaluation settings (Snoek et al., 2012; Shahriari et al., 2015). BO uses a surrogate model and acquisition function to efficiently search for high-performing hyperparameters with fewer evaluations.

Although BO is an effective approach, its application in discrete input spaces such as LoRA is limited (Oh et al., 2019; Deshwal & Doppa, 2021; Chu et al., 2024). To mitigate this, several studies (Zhang et al., 2023; Ramos et al., 2023; Agarwal et al., 2025) have shown that hybrid frameworks combining LLMs with BO represent a promising direction, achieving empirical gains across diverse domains. Such approaches include using LLM agents to propose candidate hyperparameter configurations (Liu et al., 2024a), reformulating BO tasks in natural language to flexibly incorporate search spaces and constraints (Liu et al., 2024c), and enhancing surrogate models with LLM embeddings as input features (Nguyen et al., 2024). These synergies between LLMs and BO extend beyond HPO to other domains, further emphasizing their effectiveness (Ranković & Schwaller, 2023; 2025). Building on this trend, we propose the first framework that integrates BO with LLMs for LoRA HPO. We construct an embedding space tailored to LoRA HPO using an LLM with domain prompting and learnable tokens, and perform BO within this space, improving search efficiency under high-cost evaluation conditions.

## 3. Method

We propose a framework that combines a Large Language Model (LLM) with Bayesian Optimization (BO) to discover appropriate hyperparameters for LoRA tuning. We obtain continuous embeddings from the LLM and use them as inputs to the surrogate model, enabling a BO process tailored to LoRA Hyperparameter Optimization (HPO). The LLM in our framework not only encodes rich prior knowledge through large-scale pretraining, but also provides a convenient interface for injecting additional knowledge in textual form. Furthermore, to reduce cost, we introduce proxy training evaluation, which estimates the performance of a

full-dataset model using a model trained on a subset of the data. With these components, our framework improves not only the sample efficiency of BO, but also the computational efficiency of LoRA hyperparameter optimization as a whole. Section 3.1 introduces the preliminaries of BO, Section 3.2 presents the proposed framework and its components, and Section 3.3 details our proxy training evaluation.

### 3.1. Preliminary: Bayesian optimization

Bayesian Optimization (BO) is an efficient approach for optimizing black-box functions, particularly when the evaluation cost is expensive. The goal of BO is to find the optimal input $\mathbf{x}^*$ from a candidate pool $\mathcal{X}$ that maximizes a black-box function $f$. The objective of BO can be formulated as follows:

$$\mathbf{x}^* = \arg \max_{\mathbf{x} \in \mathcal{X}} f(\mathbf{x}). \tag{1}$$

Since $f$ is hard to estimate, *surrogate model* $\hat{f}$ is used to approximate $f$. A common choice for the *surrogate model* is a Gaussian Process (GP), which can be expressed as: $\hat{f} \sim \mathcal{GP}(\mu, k_\omega)$, where $\mu$ is the mean function and $k_\omega$ denotes the kernel function with trainable hyperparameter $\omega$. For example, the Matérn 5/2 kernel is expressed as:

$$k_\omega(\mathbf{x}, \mathbf{x}') = \sigma^2 \left( 1 + \frac{\sqrt{5}d}{\ell} + \frac{5d^2}{3\ell^2} \right) \exp\left( -\frac{\sqrt{5}d}{\ell} \right), \tag{2}$$

where $\ell$ denotes the lengthscale, $\sigma^2$ denotes covariance and $d = \|\mathbf{x} - \mathbf{x}'\|_2$. Given $n$ observed points set $\mathcal{D}_n = \{(x_i, y_i)\}_{i=1}^n$, the surrogate model $\hat{f}$ is trained on observed points $D_n$. An acquisition function $a(\mathbf{x})$ is then used to select the next evaluation point $\tilde{\mathbf{x}}$ based on the posterior from the *surrogate model*:

$$\tilde{\mathbf{x}} = \arg \max_{\mathbf{x} \in \mathcal{X}} a(\mathbf{x}|\hat{f}, \mathcal{D}). \tag{3}$$

### 3.2. Proposed Framework

**Overview**. Our framework performs 4 steps in each iteration: (1) Proxy training evaluation ($Proxy$), which fine-tunes LoRA on a subset of the dataset and measures its performance; (2) Embedding extraction using the LLM; (3) Surrogate model update; and (4) Next evaluation point suggestion. For example, in the $n$-th iteration, a hyperparameter configuration $x_n$ is selected, and its benchmark performance $y_n$ is obtained through proxy training evaluation. The configuration $x_n$ is then converted into a structured template $t_n$ via domain-aware prompting. This template, together with the learnable token $\psi$, is passed into the LLM $\phi$ and projection layer $P(\cdot; \theta)$ to produce an embedding: $\mathbf{z}_n = P(\phi(t_n, \psi); \theta)$. The surrogate model, parameterized by $\omega$, is updated by maximizing the *marginal log-likelihood* using embedding $\mathbf{z}_n$ paired with the observed target $y_n$,

---

**Algorithm 1** Pseudo code for our framework

---

1: **Require:** Candidate pool $\mathcal{X}_{\text{cand}}$, observed dataset $\mathcal{D}_n = \{(x_i, y_i)\}_{i=1}^n$, budget $N$, parameters $\omega$ (GP), $\theta$ (Projection layer), $\psi$ (Learnable token), LLM $\phi$, acquisition function $a$, feature extractor $g(\cdot; \theta, \psi)$
2: **Initialize:** parameters $\omega, \theta, \psi$; $\mathcal{D}_0 \leftarrow \emptyset$; Choose initial candidate $x_1 \in \mathcal{X}_{\text{cand}}$
3: **for** $n = 1$ **to** $N$ **do**
4:     $y_n \leftarrow Proxy(x_n)$
5:     $\mathcal{D}_n \leftarrow \mathcal{D}_{n-1} \cup \{(x_n, y_n)\}$
6:     Remove $x_n$ from $\mathcal{X}_{\text{cand}}$
7:     **while** not convergence **do**
8:         **for all** $x_i \in \mathcal{D}_n$ **do**
9:             $t_i \leftarrow \text{Template}(x_i)$
10:             $\mathbf{z}_i \leftarrow g(\mathbf{x}_i; \theta, \psi) = P(\phi(t_i, \psi); \theta)$
11:         **end for**
12:         Compute $\log p(\mathbf{y}|\mathbf{Z}, \omega, \theta, \psi)$
13:         Update $\omega, \theta, \psi$
14:     **end while**
15:     **for all** $x_j \in \mathcal{X}_{\text{cand}}$ **do**
16:         $t_j \leftarrow \text{Template}(x_j)$
17:         $\mathbf{z}_j \leftarrow g(\mathbf{x}_j; \theta, \psi) = P(\phi(t_j, \psi); \theta)$
18:         Compute $a(\mathbf{z}_j; \omega, \theta, \psi)$
19:     **end for**
20:     $j' = \arg\max_j a(\mathbf{z}_j; \omega, \theta, \psi)$
21:     Suggest next evaluation point $x_{n+1} \leftarrow x_{j'}$
22: **end for**
23: $(x^*, y^*) \leftarrow \arg\max_{(x,y) \in \mathcal{D}} y$
24: **return** $x^*$

---

jointly updating all trainable parameters $\omega$, $\theta$, and $\psi$. Finally, the next evaluation point is selected by generating embeddings for every hyperparameter configuration $x_j$ in the candidate pool $\mathcal{X}_{cand}$ and evaluating the acquisition function $a(\mathbf{z})$. Algorithm 1 describes the entire procedure in pseudo-code.

**Domain-aware prompting**. We employ domain-aware prompting to explicitly incorporate domain knowledge about LoRA hyperparameters into the optimization process. A straightforward text template can be written as $t = \{\text{name}, \text{value}\}$ (Ranković & Schwaller, 2025). However, this simple format fails to capture the roles or relationships between hyperparameters. Prior studies have highlighted practical know-how and manual tuning guidelines for hyperparameter tuning (Mohammed & Kora, 2025; He, 2024; Diehl, 2024; unsloth, 2025). For example, Hu et al. (2022) suggests that the scaling factor ($\alpha$) in LoRA behaves similarly to adjusting the learning rate. To better reflect these existing know-how and guidelines, we design a structured text template $t = \{\text{explanation}, \text{name}, \text{value}\}$. The explanations include the relationship between hyperparameters (*e.g.*, how rank and alpha are commonly set) as well

as the training dynamics that arise when their magnitudes vary. This approach goes beyond simple LLM embeddings, enabling the direct insertion of structured, prompt-level information into the embedding space.

**Learnable token and projection layer**. Calibrating the embedding space extracted from the LLM during feature extraction can enhance BO performance compared to using fixed embeddings (Kristiadi et al., 2024; Ranković & Schwaller, 2025). Motivated by this insight, we introduce a learnable token $\psi$ along with a projection layer $P(\cdot; \theta)$, parameterized by $\theta$, to transform embeddings into a space better suited for BO. We append the learnable token to the domain-aware text template $t$ and feed both to the LLM to extract embeddings, allowing the token to compactly capture knowledge of LoRA hyperparameters. The embedding is obtained by pooling the hidden state at the last token position in the sequence. The embeddings are then passed through the projection layer, which generates representations tailored for BO, yielding the final feature: $\mathbf{z} = P(\phi(t, \psi); \theta)$. Throughout this process, the pre-trained LLM remains frozen, while $\psi$ and $\theta$ are learnable. As a result, the embedding not only explicitly reflects the explanations encoded in the prompt but also implicitly internalizes LoRA-specific domain knowledge. This improves representational power and enhances BO efficiency with minimal additional parameters.

**Bayesian optimization with LLM**. BO typically employs Gaussian Processes (GPs) as surrogate models, which are effective for modeling distributions over continuous spaces (Beckers, 2021). However, when dealing with complex input spaces that require understanding the relationships among variables, it becomes crucial to use representations capable of capturing such structure (Lee et al., 2025). This is particularly true for the LoRA hyperparameter space, which is inherently discrete and requires domain knowledge. To address this challenge, we integrate an LLM with a learnable token and a projection layer, which injects domain knowledge about LoRA HPO when extracting embeddings: $\mathbf{z} = g(\mathbf{x}; \theta, \psi) = P(\phi(t, \psi); \theta)$. Therefore, we employ LLM-based deep kernel learning to combine the prior knowledge encoded in the LLM with these trainable neural architectures for the GP, thereby transforming standard GP regression into deep kernel learning:

$$k(\mathbf{x}, \mathbf{x}'|\omega) \rightarrow k(g(\mathbf{x}; \theta, \psi), g(\mathbf{x}'; \theta, \psi)|\omega, \theta, \psi). \quad (4)$$

We jointly optimize all trainable parameters, $\Phi = \{\omega, \theta, \psi\}$, where $\omega$, $\theta$, and $\psi$ denote the GP kernel, projection layer, and learnable token parameters, respectively. These are optimized by maximizing the marginal log-likelihood:

$$\mathcal{L}(\Phi) = \log p(\mathbf{y}|\mathbf{X}, \Phi)$$
$$= -\frac{1}{2}\{(\mathbf{y} - \mu\mathbf{1})^\top \mathbf{K}_\Phi^{-1}(\mathbf{y} - \mu\mathbf{1}) + \log|\mathbf{K}_\Phi| + n\log 2\pi\},$$
$$(5)$$

*Table 1.* **Hyperparameter search range.** We set the hyperparameter search ranges based on prior work (Meng et al., 2024; Wang et al., 2024; Inouye et al., 2024; Diehl, 2024; Yan et al., 2025; unsloth, 2025), resulting in a search space of over 45,000 configurations.

| Hyperparameters | Search Range | Count |
|---|---|---|
| Rank ($r$) | $1 \sim 256\ (2^n)$ | 9 |
| Scaling Factor ($\alpha$) | $\frac{r}{2} \sim 128r\ (2^n r)$ | 9 |
| Batch Size | $2 \sim 256\ (2^n)$ | 8 |
| Learning Rate | $1\text{e-}6 \sim 5\text{e-}3$ | 10 |
| Dropout Rate | $0.0 \sim 0.3\ (0.05 \times n)$ | 7 |

$$\Phi^* = \arg\max_{\Phi} \mathcal{L}(\Phi), \qquad (6)$$

where $\mathbf{K}_{\Phi}$ denotes the covariance matrix determined from the covariance kernel of the GP, $\mathbf{X} = \{\mathbf{x}_1, \mathbf{x}_2, ..., \mathbf{x}_n\}$ and $\mathbf{y} = \{y_1, y_2, ..., y_n\}$.

### 3.3. Proxy training evaluation

Previous studies (Klein et al., 2017; Oliver & Wang, 2024) have shown that it is not always necessary to train on the entire dataset at every optimization step because training performance on subset datasets strongly correlates with that of full training. Building on these insights, we introduce a proxy training evaluation strategy to reduce fine-tuning time cost. Specifically, instead of training on the full dataset, we fine-tune the model on a randomly selected subset and measure performance on this smaller training run as a proxy for the true performance. Despite its simplicity, this approach exhibits strong correlation with the true performance, and we find that using only 10% of the data can be sufficient. As a result, we reduce the overall time cost by up to 10x, enabling more optimization iterations within the same computational budget. Empirically, we observe that our simple random subset achieves comparably high correlation with full-data results relative to more sophisticated data selection strategies such as Liu et al. (2024d) (Sec. 4.2).

## 4. Experiments

### 4.1. Experimental Setting

**LoRA hyperparameters and setup**. We define the candidate pool of LoRA hyperparameters as shown in Table 1. Specifically, we optimize five hyperparameters: rank ($r$), scaling factor ($\alpha$), learning rate, dropout rate, and batch size We set the search range for hyperparameters based on existing literature. Typically, the rank ($r$), scaling factor ($\alpha$), and batch size are chosen as powers of 2, while the learning rate is selected from the values considered in various LoRA variants. The dropout rate is set with a step size of 0.05, ranging from 0.0 to 0.3. As a result, we consider over 45,000 configurations. To validate our proposed framework,

we conduct experiments across multiple LoRA variants, including rsLoRA (Kalajdzievski, 2023), DoRA (Liu et al., 2024b), and PiSSA (Meng et al., 2024).

**Tasks**. Following prior work (Meng et al., 2024), we conduct experiments on three tasks: math reasoning, code generation, and conversation. For math reasoning, we fine-tune models on the MetaMathQA dataset (Yu et al., 2023) and evaluate performance on GSM8k (Cobbe et al., 2021) and MATH (Hendrycks et al., 2021), reporting Accuracy (%). For code generation, we fine-tune on the CodeFeedback dataset (Zheng et al., 2024), and evaluate on HumanEval (Chen et al., 2021) and MBPP (Austin et al., 2021), reporting Pass@1, which is the probability that the first generated solution solves the task. Finally, for conversation, we fine-tune on the WizardLM-Evol-Instruct (Xu et al., 2023) and evaluate on MT-Bench (Zheng et al., 2023). Each training dataset contains 100K samples, with a 10K subset used for proxy training evaluation.

**Baselines**. We benchmark our framework against several HPO methods: random search (Bergstra & Bengio, 2012), Optuna (Akiba et al., 2019), standard bayesian optimization (BO) (Oliver & Wang, 2024), latent bayesian optimization (LBO) (Li et al., 2021), and NOMAD (Tribes et al., 2024). To ensure fairness, all methods are constrained to 30 iterations. Details for the implementation are in the Appendix.

### 4.2. Experimental Results

**Hyperparameter optimization for LoRA variants and various models**. Based on previous findings that tasks and architectures demand distinct hyperparameters (Sengupta et al., 2024; He, 2024; Mohammed & Kora, 2025), we evaluate our framework across diverse LoRA variants and models. Table 2 shows that adapting our HPO framework enables effective hyperparameter search for each LoRA variant, consistently improving performance compared to the originally reported results. Surprisingly, our framework achieves up to 21.46% accuracy improvement, emphasizing the importance of hyperparameter selection. These results suggest that there is significant room for improvement in existing LoRA variants through systematic hyperparameter search. Similarly, Table 3 demonstrates that our approach can identify appropriate model-specific hyperparameters. Across different backbone Large Language Models (LLMs), adapting our framework consistently achieves substantial improvements, highlighting its practical utility for fine-tuning newly introduced models. Overall, our framework can be applied as a plug-and-play module that adapts to the LoRA variants and models, giving consistent gains.

We further analyze the hyperparameter combinations discovered in our experiments. Smaller batch sizes are often preferred, consistent with prior findings (Marek et al., 2025).

*Table 2.* **Results of applying our framework to LoRA variants.** We set the hyperparameter configurations suggested by each work, where they were dedicatedly tuned (Kalajdzievski, 2023; Liu et al., 2024b; Meng et al., 2024; Wang et al., 2024). Using our method, we observe consistent performance improvements across all variants.

| Strategy | Ours | Accuracy (%) | | Pass@1 | | GPT-Score |
|---|---|---|---|---|---|---|
| | | GSM8K | MATH | HumanEval | MBPP | MT-Bench |
| LoRA (Hu et al., 2022) | ✗ | 41.47 | 5.24 | 16.31 | 35.47 | 7.181 |
| | ✓ | 62.93 (+21.46) | 12.88 (+7.64) | 30.49 (+14.18) | 42.59 (+7.12) | 7.350 (+0.169) |
| rsLoRA (Kalajdzievski, 2023) | ✗ | 41.16 | 5.46 | 16.46 | 35.72 | 7.300 |
| | ✓ | 58.15 (+16.99) | 10.76 (+5.30) | 29.87 (+13.41) | 42.06 (+6.34) | 7.662 (+0.362) |
| DoRA (Liu et al., 2024b) | ✗ | 40.11 | 5.36 | 17.07 | 36.51 | 7.125 |
| | ✓ | 57.01 (+16.90) | 10.78 (+5.42) | 30.58 (+13.51) | 42.33 (+5.82) | 7.475 (+0.350) |
| PiSSA (Meng et al., 2024) | ✗ | 52.46 | 7.34 | 22.56 | 40.48 | 7.200 |
| | ✓ | 60.88 (+8.42) | 12.06 (+4.72) | 31.71 (+9.15) | 41.53 (+1.05) | 7.475 (+0.275) |

*Table 3.* **Results of applying our framework across diverse models.** We compare against the hyperparameter settings suggested by PiSSA (Meng et al., 2024), where they were dedicatedly tuned. The experiments demonstrate that our method is effective across a wide range of models.

| Model | Ours | Accuracy (%) | | Pass@1 | | GPT-Score |
|---|---|---|---|---|---|---|
| | | GSM8K | MATH | HumanEval | MBPP | MT-Bench |
| LLaMA2-7B (Touvron et al., 2023) | ✗ | 41.47 | 5.24 | 16.31 | 35.47 | 7.181 |
| | ✓ | 62.93 (+21.46) | 12.88 (+7.64) | 30.49 (+14.18) | 42.59 (+7.12) | 7.350 (+0.169) |
| Mistral-7B-v0.1 (Jiang et al., 2023) | ✗ | 69.90 | 19.96 | 45.73 | 61.90 | 8.425 |
| | ✓ | 74.07 (+4.17) | 23.46 (+3.5) | 54.27 (+8.54) | 65.08 (+3.18) | 8.688 (+0.263) |
| Gemma-7B (Team et al., 2024) | ✗ | 75.51 | 29.44 | 49.39 | 63.23 | 8.363 |
| | ✓ | 78.77 (+3.26) | 30.24 (+0.8) | 53.05 (+3.66) | 67.46 (+4.23) | 8.488 (+0.125) |

*Table 4.* **Comparison against existing HPO methods.** Our method outperforms existing HPO approaches under the same optimization budget.

| Search Method | Accuracy (%) | | Pass@1 | |
|---|---|---|---|---|
| | GSM8K | MATH | HumanEval | MBPP |
| Random | 59.14 | 10.51 | 23.17 | 36.77 |
| Optuna | 54.13 | 10.50 | 27.44 | 38.62 |
| BO | 57.32 | 11.42 | 20.12 | 35.19 |
| LBO | 59.51 | 11.88 | 26.83 | 37.83 |
| Ours | **62.93** | **12.88** | **30.49** | **42.59** |

In addition, applying dropout often leads to better performance. Interestingly, we sometimes observe strong performance when the scaling factor ($\alpha$) is 16 or even 32 times larger than the rank. This observation has not been reported in prior studies, where $\alpha$ was set to twice the rank according to existing guidelines (Diehl, 2024; unsloth, 2025), or determined based on a rank or fixed $\alpha$ (Kalajdzievski, 2023; Sun et al., 2024; Liu et al., 2025). This suggests that there may exist settings beyond the commonly chosen rank and $\alpha$ values that can further improve performance, thereby hinting at the possibility of proposing a new guideline. We report detail result in Appendix B.

**Comparison with various HPO methods**. We evaluate the effectiveness of our framework against widely adopted HPO baselines by training LoRA with them. Table 4 summarizes the results of applying each method under the same

optimization budget. The results demonstrate that our approach identifies more suitable hyperparameters within a constrained budget. Notably, our method discovers better configurations than other BO-based methods, indicating that leveraging LLMs to provide domain knowledge about the search space can improve both search efficiency and effectiveness. We also compare our framework with Tribes et al. (2024), a dedicated approach for LoRA HPO that employs validation loss with the NOMAD algorithm for hyperparameter estimation. As shown in Table 5, our method finds more appropriate configurations in a shorter time. These experiments support that the combination of LLM and BO leads to improvement in both accuracy and efficiency.

**Ablation studies**. Our framework incorporates domain knowledge into the optimization process through three components: (1) domain-aware prompting for explicit knowledge injection, and (2) a projection layer with (3) a learnable token for implicitly encoding domain knowledge. We conduct an ablation study to evaluate the effect of each proposed component. As shown in Table 6, adding each component consistently helps BO to discover better-performing hyperparameter settings, demonstrating the effectiveness of each component. Notably, domain-aware prompting plays a crucial role in performance improvement, emphasizing the importance of explicitly injecting domain knowledge. We further analyze the differences in the optimization process introduced by each component. Without any components,

*Table 5.* **Comparison against existing LoRA HPO method.** We compare our approach with (Tribes et al., 2024), which applies the NOMAD algorithm specifically for LoRA hyperparameter tuning. Our method is both more time-efficient and more effective, achieving superior performance by a significant margin. Note that H denotes hours.

| Method | Time | GSM8K | MATH | HumanEval | MBPP |
|---|---|---|---|---|---|
| Tribes et al. (2024) | 180 H | 52.16 | 9.12 | 24.39 | 37.30 |
| Ours | 24 H | 62.93 | 12.88 | 30.49 | 42.59 |

*Table 6.* **Ablation results.** We validate each of our proposed components and find that all contribute effectively to LoRA HPO.

| Projection Layer | Domain-aware Prompting | Learnable Token | GSM8K | MATH |
|---|---|---|---|---|
| ✗ | ✗ | ✗ | 47.76 | 8.72 |
| ✓ | ✗ | ✗ | 53.98 | 9.16 |
| ✓ | ✓ | ✗ | 61.41 | 12.46 |
| ✓ | ✓ | ✓ | **62.93** | **12.88** |

BO tends to keep the learning rate nearly fixed, resulting in insufficient exploration of the search space. In contrast, BO with all components explores broadly across the hyperparameter candidate pool. These findings show that our framework enables effective exploration of diverse hyperparameters even with a small number of iterations, allowing BO to operate over a much broader search space.

**Qualitative analysis of the effect of our framework**. We visualize the embedding $z$ of hyperparameter configurations to illustrate the effect of adding each component of our framework, as shown in Figure 1. The figure compares three settings: (a) frozen LLM embeddings, (b) embeddings after applying the projection layer, and (c) embeddings with both the projection layer and a learnable token. With frozen LLM embeddings, high- and low-performing hyperparameter configurations remain entangled, leading to an unstable search process. These results indicate that the embedding space does not effectively separate hyperparameter combinations, potentially hindering the balance between exploration and exploitation. Introducing a projection layer begins to separate the embeddings, revealing clearer structures that distinguish the performance levels. When we additionally incorporate a learnable token, the embeddings exhibit directional organization aligned with performance, enabling more reliable surrogate fitting and a better-organized space overall. Furthermore, we analyze the trajectories of the BO process across different settings and find that optimization using only a frozen LLM proceeds without a clear direction. In contrast, when components for embedding calibration are included, the BO process consistently moves toward the high-performing region. These observations suggest that calibration with a projection layer and a learnable token makes the BO landscape more discriminative and smoother

*Table 7.* **Correlation between the performance trained on a subset and on the full dataset.** Proxy training evaluation shows comparable correlation to full dataset accuracy, at both random sampling and TSDS (Liu et al., 2024d).

| Sampling Method | MATH Reasoning | Code Generation |
|---|---|---|
| Random (1%) | 0.7031 | 0.7429 |
| Random (5%) | 0.8360 | 0.9282 |
| Random (10%) | 0.8713 | **0.9427** |
| TSDS (10%) by Test dataset | **0.8754** | 0.9290 |
| TSDS (10%) by Train dataset | 0.8649 | 0.9278 |

*Table 8.* **Performance differences across model sizes.** We apply the hyperparameters discovered for each model size of Qwen2.5 to fine-tune all model sizes. "Model" denotes the model being fine-tuned, while "Settings" indicates the size of the model from which the hyperparameters were obtained.

| Model | Settings | Accuracy (%) | | Pass@1 | |
|---|---|---|---|---|---|
| | | GSM8K | MATH | HumanEval | MBPP |
| Qwen2.5-3B | 3B | 79.53 | 43.18 | 70.12 | 77.78 |
| | 7B | 78.54 | 42.40 | 68.29 | 75.13 |
| | 14B | 78.47 | 42.94 | 67.07 | 77.25 |
| Qwen2.5-7B | 3B | 84.08 | 48.90 | 81.09 | 78.84 |
| | 7B | 83.93 | 48.08 | 79.88 | 78.57 |
| | 14B | 83.09 | 48.58 | 81.71 | 78.31 |
| Qwen2.5-14B | 3B | 87.41 | 51.68 | 82.32 | 82.80 |
| | 7B | 86.81 | 50.62 | 79.88 | 81.22 |
| | 14B | 87.34 | 51.50 | 81.71 | 82.01 |

than using fixed embeddings, thereby improving search efficiency and final performance.

**Analysis of embedding geometry**. To further extend the qualitative analysis, we investigate what information is encoded in the LLM embeddings after applying our framework. Specifically, as shown in Figure 2, we apply Principal Component Analysis (PCA) to the learned embeddings and sort configurations along the first principal component in the LLaMA2-7B setting. We then compare this ordering with the surrogate-predicted performance and observe a strong monotonic relationship among the top 300 configurations (*i.e.*, Spearman's $\rho = 0.8779$). This indicates that the dominant direction of the calibrated embedding space closely aligns with the optimization objective, providing a continuous ordering of configurations by predicted quality. Moreover, configurations located in the promising region along this axis exhibit consistent hyperparameter patterns, including larger ranks, smaller batch sizes, learning rates around $1 \times 10^{-4}$, and non-zero dropout, which are consistent with the observations above. In conclusion, this analysis suggests that the embeddings calibrated by our framework can reshape the embedding geometry into a more suitable space for BO, where hyperparameter configurations are coherently organized with respect to the optimization target.

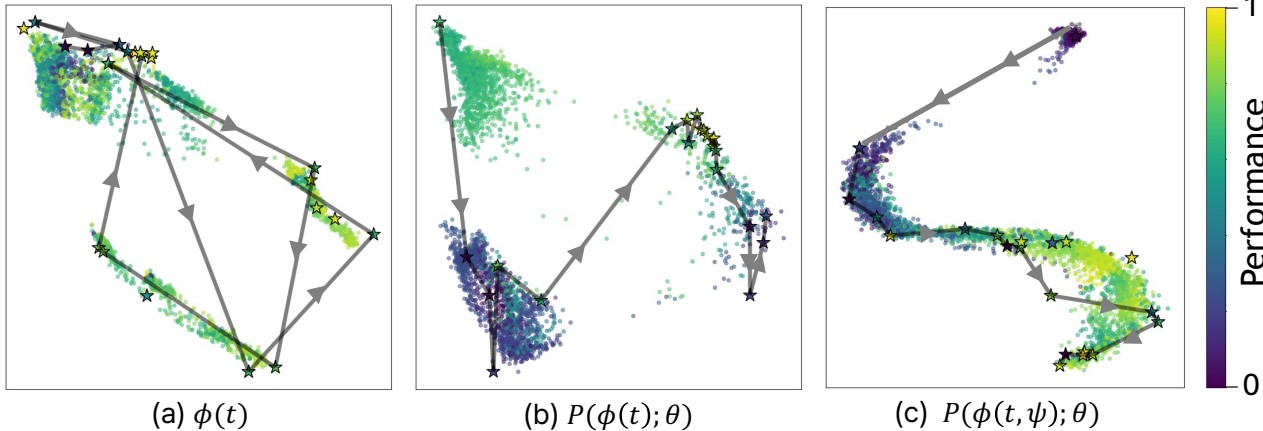

*Figure 1.* **Qualitative ablation study of our components.** We visualize how the embedding space evolves with our proposed components: (a) shows the embedding space from a frozen LLM $\phi$; (b) shows the space when a projection layer $P(\cdot; \theta)$ is added to the frozen LLM; and (c) shows the space when both the projection layer and the learnable token $\psi$ are employed. Dots are all possible points, and stars indicate visited points during the optimization. The trajectories in each figure indicate the main paths across steps, shown in arrow sequence. For clarity, only a few selected optimization trajectories are shown in each figure. These results suggest that incorporating the projection layer and learnable token produces a smoother, more structured embedding space, thereby enabling efficient optimization.

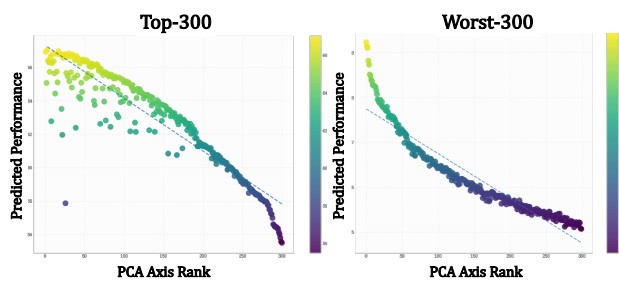

*Figure 2.* **Analysis of embedding geometry.** We provide PCA visualization of the calibrated LLM embeddings. The color bar represents the performance estimated by the surrogate model.

**Validation of proxy training evaluation**. To reduce the time cost of fine-tuning during HPO, we introduce proxy training evaluation in Section 3.3. Using the proposed proxy training evaluation, we estimate performance by training on a subset of the dataset and treating it as a proxy for full-data performance. To further investigate this, we examine the Pearson correlation between the training performance of subset datasets at various sampling ratios and that of the full dataset. Additionally, we compare with the existing data sampling method, TSDS (Liu et al., 2024d), setting target distribution to either the test or the training dataset. Table 7 demonstrates that proxy training evaluation with a randomly selected 10% subset provides a sufficiently accurate approximation of full dataset performance. These results indicate that our proxy training evaluation serves as an effective and reliable indicator of model performance on the full dataset. Moreover, the correlation obtained from the 10% random subset is comparable to that of TSDS (Liu et al., 2024d) and even achieves the highest correlation in the code generation task. Based on these findings, we adopt

*Table 9.* **Results of cross-applying hyperparameters across models.** We observe performance degradation when hyperparameters discovered for one model series are applied to another. This indicates that our framework effectively searches for hyperparameters suited to each model.

| Model | Settings | GSM8K | MATH |
|---|---|---|---|
| LLaMA2-7B | LLaMA2-7B | 62.93 | 12.88 |
| | Qwen2.5-3B | 39.5 | 5.2 |
| | Qwen2.5-7B | 32.68 | 4.7 |
| | Qwen2.5-14B | 52.46 | 8.06 |
| Qwen2.5-7B | Qwen2.5-7B | 83.93 | 48.08 |
| | LLaMA2-7B | 81.12 | 41.06 |
| | LLaMA2-13B | 80.06 | 40.18 |

10% random sampling to construct the subset dataset.

**Effect of model size on LoRA HPO**. We investigate the effect of model size on the selection of suitable LoRA hyperparameters. Specifically, we apply our framework to Qwen2.5 models with 3B, 7B, and 14B parameters, identifying the best hyperparameters for each model scale. We then use these configurations to fine-tune models across all scales. As shown in Table 8, our framework consistently performs well across different Qwen2.5 model scales. Notably, configurations discovered at one scale remain effective when transferred to other scales within the same model family. Since the configurations are largely transferable across scales within the same model series, this implies that tuning costs can be reduced by applying configurations found on smaller models to larger ones, rather than running the framework directly on larger models. However, while configurations found on other scales remain effective, scale-matched configurations tend to yield the highest performance. In

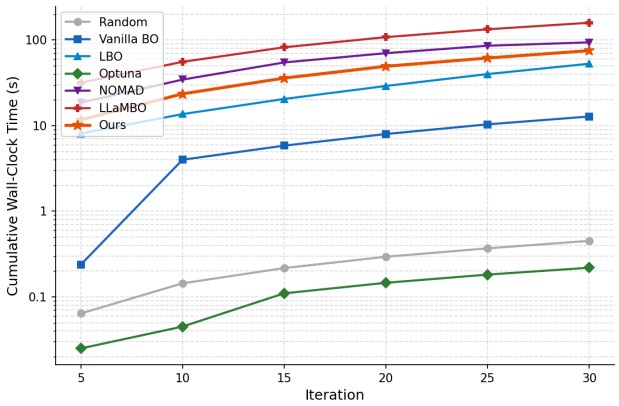

*Figure 3.* **Efficiency curve.** We compare the efficiency of various HPO methods by measuring cumulative time. We observe that our method is sufficiently reasonable and practical when considering the trade-off with performance.

contrast, we observe clear differences between model architectures. Compared to LLaMA2, Qwen2.5 models tend to prefer smaller ranks and larger batch sizes. This architectural dependency becomes more evident in Table 9. When configurations found on Qwen2.5 are applied to LLaMA2 models, and vice versa, we observe large performance degradation than when each configuration is used within the same model series. These observations indicate that LoRA hyperparameters are more influenced by model architecture than by scale, consistent with prior findings (Yan et al., 2025).

**Efficiency comparison with various HPO methods**. To compare the efficiency of the proposed framework with other HPO methods, we measure the cumulative wall-clock time at each iteration. In this experiment, we exclude the proxy training time required to obtain the performance of LoRA fine-tuning, and only consider the optimization time spent at each iteration. As shown in Figure 3, our framework identifies effective hyperparameter configurations with substantially less time than LLAMBO (Liu et al., 2024c), a BO method leveraging LLMs, and NOMAD (Tribes et al., 2024), which is specialized for LoRA hyperparameter search. We also observe that the time difference between our framework and Latent BO remains marginal. Although Optuna and random search require less time, the results demonstrate that our framework provides a reasonable and practical pipeline when considering the trade-off between optimization cost and final performance.

**Generalizability to broader HPO tasks**. To further investigate whether our LLM-based BO framework with domain-aware prompting and learnable tokens can be extended beyond LoRA HPO, we conduct additional experiments on general HPO tasks in Bayesmark (Uber, 2020). Figure 4 reports the results on three representative models: Decision Tree (DT), Multi-Layer Perceptron (MLP), and Random Forest (RF), across two classification tasks (*i.e.*, breast cancer

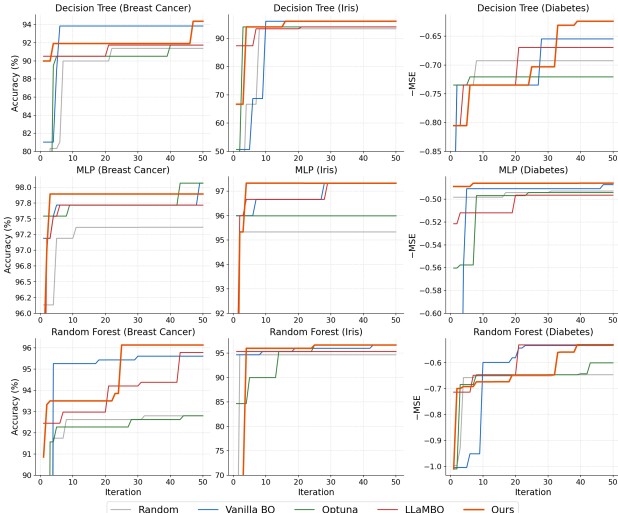

*Figure 4.* **Generalizability experiments on Bayesmark.** To examine the extensibility of our framework, we report experimental results on a standard HPO benchmark commonly used in machine learning. By achieving the best performance in most experimental settings, our method demonstrates its applicability.

and IRIS) and one regression task (*i.e.*, diabetes). For each model, we construct domain-aware prompts using simple textual descriptions of the hyperparameters being optimized. The results show that our proposed method achieves the best or highly competitive performance in most settings compared to various HPO baselines. These findings demonstrate that the proposed framework remains effective even when applied to general HPO problems. This suggests that our approach has the potential to serve as a broadly applicable optimization framework for diverse HPO tasks.

## 5. Conclusion

We propose a framework that integrates Large Language Models (LLMs) with Bayesian optimization (BO) for LoRA Hyperparameter Optimization (HPO). Domain knowledge about LoRA is explicitly incorporated into the BO process through domain-aware prompting, while a learnable token and a projection layer implicitly transform LLM embeddings into a space suited for optimization. To further reduce computational cost, we employ empirically validated proxy training evaluation that estimates fine-tuning performance using a subset of the training data. As a result, our framework efficiently identifies high-performing hyperparameter configurations from a large candidate pool, significantly reducing optimization time. It operates as a plug-and-play module and achieves consistent performance improvements across different LoRA variants, model architectures, and model scales. Comparisons with existing HPO methods demonstrate its effectiveness in both cost and performance. Beyond LoRA, we expect this framework to serve as a practical baseline in diverse fine-tuning strategies.

## Acknowledgement

This work was supported by Samsung Electronics Co., Ltd (Project Code: IO240508-09825-01); the InnoCORE program (N10250156, KAIST InnoCore LLM), 25%; Institute of Information & communications Technology Planning & Evaluation (IITP) grant (No. RS-2024-00457882, National AI Research Lab Project, 25%; No. 2022-0-00124, No.RS-2022-II220124, Development of Artificial Intelligence Technology for Self-Improving Competency-Aware Learning Capabilities, 25%; No.RS-2019-II191906, Artificial Intelligence Graduate School Program (POSTECH), 25%) funded by the Korea government (MSIT).

## Impact Statement

This paper presents a novel framework combining LLMs and BO to efficiently search for LoRA hyperparameters. By leveraging domain-aware prompting and proxy training evaluation, this approach significantly reduces the computational cost of hyperparameter optimization, making it applicable not only to LoRA but also to its variants such as DoRA, rsLoRA, and PiSSA. By applying our framework, we can reduce the computational cost and time required for hyperparameter tuning. It can accelerate the development of not only LLMs but also models generally used in machine learning. Furthermore, our framework is designed for LoRA hyperparameter tuning but can also be applied to domains that require domain knowledge, such as healthcare, medical, and manufacturing processes. Therefore, our method can lay the foundation for future researchers to optimize processes specific to these fields. However, there is a potential risk of misuse in security-sensitive areas or in fields where small errors can lead to severe consequences.

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

## A. Optimizing $\omega, \theta, \psi$ through marginal log-likelihood

This part is inspired by Wilson et al. (2016) and Ranković & Schwaller (2025), from which we partially adopt several equations. We formulate *marginal log-likelihood* as follows:

$$\mathcal{L}(\Phi) = \log p(\mathbf{y}|\mathbf{X}, \Phi) = -\frac{1}{2}\{(\mathbf{y} - \mu\mathbf{1})^\top \mathbf{K}_\Phi^{-1}(\mathbf{y} - \mu\mathbf{1}) + \log |\mathbf{K}_\Phi| + n \log 2\pi\}, \tag{7}$$

where $\mathbf{X} = \{\mathbf{x}_1, \mathbf{x}_2, ..., \mathbf{x}_n\}$ and $\mathbf{y} = \{y_1, y_2, ..., y_n\}$.

To maximize *marginal log-likelihood*, gradient-based optimization is used to optimize kernel hyperparameter $\omega$, weight of projection layer $\theta$, and learnable token $\psi$. We define the parameter set $\Phi = \{\omega, \theta, \psi\}$. The gradient of the *marginal log-likelihood* can be computed by applying the chain rule with respect to each parameter, resulting in the following decomposition:

$$\frac{\partial \mathcal{L}}{\partial \boldsymbol{\omega}} = \frac{\partial \mathcal{L}}{\partial K_\Phi}\frac{\partial K_\Phi}{\partial \boldsymbol{\omega}}, \quad \frac{\partial \mathcal{L}}{\partial \theta} = \frac{\partial \mathcal{L}}{\partial K_\Phi}\frac{\partial K_\Phi}{\partial g(\mathbf{x};\theta,\psi)}\frac{\partial g(\mathbf{x};\theta,\psi)}{\partial \theta}, \quad \frac{\partial \mathcal{L}}{\partial \psi} = \frac{\partial \mathcal{L}}{\partial K_\Phi}\frac{\partial K_\Phi}{\partial g(\mathbf{x};\theta,\psi)}\frac{\partial g(\mathbf{x};\theta,\psi)}{\partial \psi}, \tag{8}$$

$$\frac{\partial \mathcal{L}}{\partial K_\Phi} = \frac{1}{2}K_\Phi^{-1}(\mathbf{y} - \mu\mathbf{1})(\mathbf{y} - \mu\mathbf{1})^\top K_\Phi^{-1} - \frac{1}{2}K_\Phi^{-1}, \tag{9}$$

where $\frac{\partial K_\Phi}{\partial \boldsymbol{\omega}}$ are the derivatives of the kernel with respect to the kernel hyperparameters, $\frac{\partial K_\Phi}{\partial g(\mathbf{x};\theta,\psi)}$ means the implicit derivatives of the kernel with respect to the $g$. $\frac{\partial g(\mathbf{x};\theta,\psi)}{\partial \theta}$ are the derivatives of the projection layer parameters via backpropagation and $\frac{\partial g(\mathbf{x};\theta,\psi)}{\partial \psi}$ are the derivatives of the learnable token parameters via backpropagation. Finally, we can compute the gradient of *marginal log-likelihood* by applying the chain rule.

## B. Details of the Experimental Setting

**Implementation details**. Motivated by Ranković & Schwaller (2025), we use Qwen2-7B as the LLM in our framework to extract embeddings for BO, applying last-token pooling. The embedding dimension is set to 3584. We define the projection layer as follows:

$$P(\mathbf{x};\theta) = \mathrm{ELU}(\mathrm{Dropout}(W\mathbf{x} + b)). \tag{10}$$

We also utilize the Matérn-5/2 kernel and Expected Improvement (EI) as an acquisition function based on previous studies (Ranković et al., 2024; Ranković & Schwaller, 2025). The backbone LLM used for experiments on LoRA variants is LLaMA2-7B. For cases where our method is not applied, we follow the settings reported in prior work (Meng et al., 2024; Wang et al., 2024). We set the model's sequence length to 1024 and use a warmup ratio of 0.03 with a cosine learning rate scheduler. To reduce computational cost, all models are trained for a single epoch. We adopt the commonly used acquisition function, Expected Improvement (EI), where $x^*$ isthe current maximum input:

$$a(x|\hat{f}, \mathcal{D}) = \mathbb{E}[I(x)] = \mathbb{E}[\max(f(x) - f(x^*), 0)] \tag{11}$$

**Details on competing methods**. We conduct our experiments on several HPO methods, random search (Bergstra & Bengio, 2012), Optuna (Akiba et al., 2019), standard BO (Oliver & Wang, 2024), latent BO (LBO) (Li et al., 2021), LLAMBO (Liu et al., 2024c) and NOMAD (Tribes et al., 2024). To ensure fairness, all methods are constrained to 30 optimization iterations. For each method, the top-1 result is obtained by selecting the best hyperparameter configuration on the training subset. We use the BoTorch library (Balandat et al., 2020) for BO and LBO implementations, conducting hyperparameter search with the same design space as ours. For LBO, we adapt the feature extractor proposed by (Lee et al., 2025), which consists of two repeated blocks of linear and ReLU layers with a hidden dimension of 64. For Optuna, we use the default TPE setting

with integer hyperparameter candidates. For the method of (Tribes et al., 2024), we run the NOMAD algorithm by executing our LoRA tuning Python script. All experiments are conducted on two A100-80GB GPUs.

**Discovered hyperparameters for each experiment**. The hyperparameters discovered after optimization and used for training are reported in Tables 10 and 12. Tables 13 present the details of the hyperparameters identified by the competing search methods, while Table 14 reports those obtained during the ablation studies. Our experiments show that, when applied to diverse models and LoRA variants, our framework consistently discovers hyperparameter configurations with a higher rank than the baselines. This suggests that our method effectively identifies hyperparameters most appropriate for each model and each LoRA variant.

## C. Additional Results

We provide supplementary experiments and analyses in addition to the main results presented in the main paper.

**Validation on models of different sizes**. Table 15 summarizes the validity of our framework under model size variations in LLaMA2. Even when the model size increases from 7B to 13B, our method successfully identifies appropriate hyperparameters, demonstrating the framework's robustness to changes in scale.

**Cross-application of hyperparameters within the same model series**. We apply the same procedure as in Table 8 in the main paper to the LLaMA2 series, transferring hyperparameter settings discovered for one model size to another. The results in Table 16 show that hyperparameter configurations can remain effective across different scales within the same series. This further suggests the possibility of searching for hyperparameters on smaller models and transferring them to larger ones.

**Cross-application of hyperparameters between different model series**. To examine whether hyperparameters identified in one model transfer to another, we conduct experiments applying configurations discovered on Qwen2.5 to LLaMA-2, and vice versa, as shown in Table 9. Applying hyperparameters found on Qwen2.5 to LLaMA-2 leads to substantial performance degradation. Similarly, applying those from LLaMA-2 to Qwen2.5 also degrades performance. These results support the claim in Sec 4.2 in the main paper that preferred hyperparameter settings vary with model architecture.

**Correlation between proxy training and full training**. Table 17 shows that the correlation between subset training and full training remains consistent across all benchmarks. Notably, randomly sampling only 10% of the data still yields high correlation with full-dataset performance. This supports the claim in Sec 4.2 in the main paper that random sampling is a reasonable and efficient choice, comparable to more sophisticated data selection methods (Liu et al., 2024d). Thus, instead of tuning on the full dataset, leveraging proxy training evaluation provides a reliable proxy for estimating model performance during hyperparameter search.

**Comparison with LLAMBO**. We present a comparative evaluation against LLAMBO (Liu et al., 2024c), a representative framework that integrates LLM with Bayesian optimization for hyperparameter search.[1] As shown in Table 18, our method consistently achieves better hyperparameter selection performance than LLAMBO, which adopts a naïve prompting-based strategy that replaces all Bayesian optimization components with an LLM. This result indicates that relying solely on an LLM's prior knowledge is insufficient for effective hyperparameter optimization. In contrast, explicitly modeling the objective with Gaussian processes and calibrating the embedding space leads to more reliable and effective optimization performance.

**Ablation on embedding model choice**. We further analyze the effect of the choice of embedding LLM. In our main experiments, we use Qwen2-7B as the default embedding model. To examine whether the effectiveness of our method depends on this specific choice, we evaluate several alternatives, including a smaller model, the same model as the inference model, and different embedding architectures such as Mistral and T5. As shown in Table 19, all embedding models improve over the default hyperparameter setting, indicating that the proposed pipeline can provide consistent gains across different embedding choices. At the same time, the final performance varies across embedding models, suggesting that the choice of embedding LLM can affect the quality of the learned hyperparameter representation. We leave a more systematic investigation of which embedding properties (*e.g.*, model scale, architecture, or hidden-state dimensionality) are most beneficial to future work.

---

[1]We have used the default setting of LLAMBO (10 MC predictions for each iteration) for the experiment with 30 iterations.

**Direct optimization on holdout dataset**. Our main experiments follow the setup of prior work on LoRA variants such as PiSSA (Meng et al., 2024), where hyperparameters are optimized on a source dataset and evaluated on related datasets. To further verify that the reported gains are not artifacts of this transfer-based evaluation protocol, we additionally apply our framework directly on the holdout dataset, GSM8K. For Qwen2.5-7B, we follow the default setting from prior work (Rathore et al., 2025). As shown in Table 20, our method outperforms the default hyperparameter setting for both models. These results suggest that the improvements are not solely due to favorable transfer from the source dataset, but also reflect the effectiveness of our HPO procedure when applied directly on the target evaluation dataset.

**Effect of automatic domain-aware prompting**. To account for scenarios where manually curated domain knowledge is unavailable, we further examine whether our framework can operate without manually specified domain knowledge. We introduce an automatic alternative to domain-aware prompting, where the surrogate model first learns the relationship between hyperparameters and performance using only evaluation feedback from the optimization loop. The learned trends are then converted into textual descriptions by using GPT, and these textual descriptions are used as automatically generated domain-aware prompts. As shown in Table 21, this alternative improves over both the default setting and our method without domain-aware prompting. This result indicates that useful hyperparameter structure can be extracted from evaluation metrics alone. Meanwhile, the manually designed domain-aware prompting achieves the best overall performance, suggesting that human-processed knowledge can serve as an effective inductive bias when available. In conclusion, these results suggest that the automatic variant can serve as a useful alternative when manually curated domain knowledge is difficult to access.

## D. Template for Domain-aware prompting

We provide an example of the template for domain-aware prompting in Table 22. This template focuses on the roles and relationships of each hyperparameter and describes how training dynamics change as their values vary, based on prior studies (Kalajdzievski, 2023; Sun et al., 2024; Diehl, 2024; Meng et al., 2024; unsloth, 2025; Liu et al., 2025). Compared to a template without domain-aware prompting, this design captures rich domain knowledge about LoRA hyperparameters, significantly improving the effectiveness of the following Bayesian Optimization. The template can also be modified.

*Table 10.* **Hyperparameters for math reasoning tasks.** We present the hyperparameter configuration used to train MetaMathQA for evaluating on the GSM8K and MATH datasets.

| Models | Strategy | Ours | Hyperparameter | | | | |
| --- | --- | --- | --- | --- | --- | --- | --- |
| | | | Rank($r$) | Scaling Factor($\alpha$) | Dropout | Batch Size | Learning Rate |
| LLaMA2-7B | LoRA | ✗ | 8 | 16 | 0.0 | 32 | 2e-05 |
| | | ✓ | 256 | 8192 | 0.0 | 4 | 5e-06 |
| | rsLoRA | ✗ | 8 | 16 | 0.0 | 32 | 2e-05 |
| | | ✓ | 128 | 1024 | 0.05 | 64 | 5e-05 |
| | DoRA | ✗ | 8 | 16 | 0.0 | 32 | 2e-05 |
| | | ✓ | 16 | 16 | 0.3 | 16 | 5e-04 |
| | PiSSA | ✗ | 128 | 128 | 0.0 | 128 | 2e-05 |
| | | ✓ | 256 | 4096 | 0.0 | 4 | 5e-06 |
| LLaMA2-13B | LoRA | ✗ | 8 | 16 | 0.0 | 32 | 2e-05 |
| | | ✓ | 32 | 512 | 0.0 | 2 | 5e-05 |
| Mistral-7B-v0.1 | LoRA | ✗ | 128 | 128 | 0.0 | 128 | 2e-05 |
| | | ✓ | 128 | 128 | 0.1 | 4 | 3e-05 |
| Gemma-7B | LoRA | ✗ | 128 | 128 | 0.0 | 128 | 2e-05 |
| | | ✓ | 64 | 256 | 0.0 | 2 | 5e-06 |
| Qwen2.5-3B | LoRA | ✓ | 1 | 4 | 0.25 | 32 | 5e-05 |
| Qwen2.5-7B | LoRA | ✓ | 32 | 64 | 0.25 | 16 | 5e-05 |
| Qwen2.5-14B | LoRA | ✓ | 1 | 4 | 0.25 | 32 | 2e-05 |

*Table 11.* **Hyperparameters for code generation tasks.** We present the hyperparameter configuration used to train CodeFeedback for evaluating on the HumanEval and MBPP datasets.

| Models | Strategy | Ours | Hyperparameter | | | | |
| --- | --- | --- | --- | --- | --- | --- | --- |
| | | | Rank($r$) | Scaling Factor($\alpha$) | Dropout | Batch Size | Learning Rate |
| LLaMA2-7B | LoRA | ✗ | 8 | 16 | 0.0 | 32 | 2e-05 |
| | | ✓ | 256 | 128 | 0.0 | 4 | 5e-05 |
| | rsLoRA | ✗ | 8 | 16 | 0.0 | 32 | 2e-05 |
| | | ✓ | 256 | 128 | 0.25 | 4 | 5e-05 |
| | DoRA | ✗ | 8 | 16 | 0.0 | 32 | 2e-05 |
| | | ✓ | 128 | 256 | 0.15 | 2 | 3e-05 |
| | PiSSA | ✗ | 128 | 128 | 0.0 | 128 | 2e-05 |
| | | ✓ | 32 | 1024 | 0.0 | 2 | 3e-05 |
| LLaMA2-13B | LoRA | ✗ | 8 | 16 | 0.0 | 32 | 2e-05 |
| | | ✓ | 256 | 128 | 0.25 | 2 | 1e-04 |
| Mistral-7B-v0.1 | LoRA | ✗ | 128 | 128 | 0.0 | 128 | 2e-05 |
| | | ✓ | 128 | 256 | 0.0 | 2 | 5e-06 |
| Gemma-7B | LoRA | ✗ | 128 | 128 | 0.0 | 128 | 2e-05 |
| | | ✓ | 256 | 256 | 0.25 | 32 | 2e-05 |
| Qwen2.5-3B | LoRA | ✓ | 128 | 64 | 0.1 | 128 | 5e-06 |
| Qwen2.5-7B | LoRA | ✓ | 8 | 4 | 0.0 | 64 | 2e-05 |
| Qwen2.5-14B | LoRA | ✓ | 128 | 128 | 0.15 | 64 | 5e-06 |

*Table 12.* **Hyperparameters for conversation tasks.** We present the hyperparameter configuration used to train WizardLM-Evol-Instruct for evaluating on the MT-Bench.

| Models | Strategy | Ours | Hyperparameter | | | | |
|---|---|---|---|---|---|---|---|
| | | | Rank($r$) | Scaling Factor($\alpha$) | Dropout | Batch Size | Learning Rate |
| LLaMA2-7B | LoRA | ✗ | 8 | 16 | 0.0 | 32 | 2e-05 |
| | | ✓ | 64 | 1024 | 0.0 | 2 | 3e-05 |
| | rsLoRA | ✗ | 8 | 16 | 0.0 | 32 | 2e-05 |
| | | ✓ | 16 | 16 | 0.2 | 2 | 2e-05 |
| | DoRA | ✗ | 8 | 16 | 0.0 | 32 | 2e-05 |
| | | ✓ | 64 | 1024 | 0.0 | 2 | 5e-06 |
| | PiSSA | ✗ | 128 | 128 | 0.0 | 128 | 2e-05 |
| | | ✓ | 16 | 1024 | 0.0 | 2 | 5e-06 |
| Mistral-7B-v0.1 | LoRA | ✗ | 128 | 128 | 0.0 | 128 | 2e-05 |
| | | ✓ | 256 | 256 | 0.25 | 64 | 5e-06 |
| Gemma-7B | LoRA | ✗ | 128 | 128 | 0.0 | 128 | 2e-05 |
| | | ✓ | 2 | 2 | 0.3 | 2 | 3e-05 |

*Table 13.* **Discovered hyperparameters by competing methods for math reasoning tasks and code generation tasks.** We present the hyperparameter configurations obtained from competing methods. The reported hyperparameters include those used to train the LLaMA2-7B model on MetaMathQA and evaluate it on the GSM8K and MATH datasets, as well as those used to train CodeFeedback and evaluate it on the HumanEval and MBPP datasets.

| Search Method | Hyperparameter (Training dataset: MetaMathQA) | | | | | Hyperparameter (Training dataset:CodeFeedback) | | | | |
|---|---|---|---|---|---|---|---|---|---|---|
| | Rank($r$) | Scaling Factor($\alpha$) | Dropout | Batch Size | Learning Rate | Rank($r$) | Scaling Factor($\alpha$) | Dropout | Batch Size | Learning Rate |
| Random | 128 | 1024 | 0.1 | 16 | 5e-05 | 4 | 8 | 0.0 | 16 | 5e-05 |
| Optuna (TPE) | 32 | 1024 | 0.1 | 16 | 1e-04 | 256 | 128 | 0.2 | 16 | 1e-04 |
| BO | 16 | 64 | 0.25 | 2 | 1e-04 | 16 | 8 | 0.15 | 2 | 5e-06 |
| LBO | 128 | 4096 | 0.0 | 2 | 5e-06 | 256 | 128 | 0.3 | 256 | 6e-04 |
| Tribes et al. (2024) | 8 | 256 | 0.1 | 4 | 1e-04 | 4 | 64 | 0.0 | 4 | 3e-05 |

*Table 14.* **Discovered hyperparameters in ablation studies.** We present the hyperparameter configuration during our ablation studies, used in Table 6 in the main paper.

| Projection Layer | Domain-aware Prompting | Learnable Token | Hyperparameter | | | | |
|---|---|---|---|---|---|---|---|
| | | | Rank($r$) | Scaling Factor($\alpha$) | Dropout | Batch Size | Learning Rate |
| ✗ | ✗ | ✗ | 64 | 32 | 0.25 | 2 | 4e-04 |
| ✓ | ✗ | ✗ | 8 | 8 | 0.1 | 8 | 1e-04 |
| ✓ | ✓ | ✗ | 128 | 256 | 0.1 | 32 | 3e-04 |
| ✓ | ✓ | ✓ | 256 | 8192 | 0.0 | 4 | 5e-06 |

*Table 15.* **Performance across different model sizes.** Adapting our framework to different model sizes consistently shows improvements, indicating its effectiveness.

| Models | Ours | Accuracy (%) | | Pass@1 | |
|---|---|---|---|---|---|
| | | GSM8K | MATH | HumanEval | MBPP |
| LLaMA2-7B | ✗ | 41.47 | 5.24 | 16.31 | 35.47 |
| | ✓ | 62.93 | 12.88 | 30.49 | 42.59 |
| LLaMA2-13B | ✗ | 55.34 | 8.68 | 29.88 | 46.56 |
| | ✓ | 64.44 | 14.68 | 42.07 | 53.17 |

*Table 16.* **Performance differences across model sizes.** We apply the hyperparameters discovered for each model size of LLaMA2 to fine-tune all model sizes.

| Model | Settings | Accuracy (%) | | Pass@1 | |
|---|---|---|---|---|---|
| | | GSM8K | MATH | HumanEval | MBPP |
| LLaMA2-7B | 7B | 62.93 | 12.88 | 30.49 | 42.59 |
| | 13B | 60.12 | 10.74 | 34.15 | 44.97 |
| LLaMA2-13B | 7B | 66.57 | 15.24 | 42.68 | 51.59 |
| | 13B | 64.44 | 14.68 | 42.07 | 53.17 |

*Table 17.* **Correlation between performance on a subset and the full dataset.** The percentages indicate the sampling ratios from the full dataset. For TSDS (Liu et al., 2024d), we report results separately when the target distribution is matched to the test dataset or the training dataset. Pearson correlation is used as the evaluation metric.

| Sampling Method | Math Reasoning | | Code Generation | |
|---|---|---|---|---|
| | GSM8K | MATH | HumanEval | MBPP |
| Random (1%) | 0.6879 | 0.4335 | 0.8052 | 0.5469 |
| Random (5%) | 0.8197 | 0.6483 | 0.8857 | 0.8879 |
| Random (10%) | 0.8566 | 0.6578 | 0.8652 | 0.9286 |
| TSDS (10%) by Test dataset | 0.8651 | 0.7117 | 0.8589 | 0.8209 |
| TSDS (10%) by Train dataset | 0.8529 | 0.6602 | 0.8624 | 0.9245 |

*Table 18.* **Comparison with LLAMBO.** We report the comparison result of LLAMBO and ours at GSM8k, MATH. As shown, our method outperforms LLAMBO, which utilizes a prompting-based method for bayesian optimization.

| Method | Accuracy (%) | |
|---|---|---|
| | GSM8K | MATH |
| LLAMBO (Liu et al., 2024c) | 48.82 | 8.76 |
| Ours | **62.93** | **12.88** |

*Table 19.* **Ablation on embedding model choice.** Different embedding models consistently improve over the default hyperparameter setting.

| Inference Model | Embedding Model | GSM8K | MATH |
|---|---|---|---|
| LLaMA2-7B | Qwen2-7B (Default) | 62.93 | 12.88 |
| | Qwen2-1.5B | 54.21 | 9.82 |
| | LLaMA2-7B | 56.49 | 10.96 |
| | Mistral-7B | 57.62 | 9.96 |
| | T5-base | 55.27 | 10.58 |
| Qwen2.5-7B | Qwen2-7B (Default) | 83.93 | 48.08 |
| | Qwen2.5-7B | 84.38 | 49.00 |

*Table 20.* **Direct optimization on holdout dataset.** Our method improves over the default hyperparameter setting even when directly applied to the holdout dataset.

| Model | Ours | GSM8K |
|---|---|---|
| LLaMA2 | ✗ | 15.54 |
| | ✓ | 39.42 |
| Qwen2.5 | ✗ | 79.08 |
| | ✓ | 83.47 |

*Table 21.* **Effect of automatic domain-aware prompting.** The automatic variant improves over the setting without domain-aware prompting, suggesting that useful structure can be extracted from optimization feedback alone. "DAP" means domain-aware prompting.

| Ours | DAP | Manual | GSM8K | MATH |
|---|---|---|---|---|
| ✗ | ✗ | – | 47.76 | 8.72 |
| ✓ | ✗ | – | 53.98 | 9.16 |
| ✓ | ✓ | ✓ | 61.41 | 12.46 |
| ✓ | ✓ | ✗ | 55.73 | 10.48 |

*Table 22.* **Prompt templates with and without domain-aware prompting.**

```
rank(r)={rank_value}, Scaling factor(α)={alpha_value}, Dropout Rate={dropout_value},
Batch Size={batchsize_value}, Learning Rate={lr_value}
```

*(a)* Prompt templates without domain-aware prompting

```
* Rank (r): Controls adapter capacity by setting the low-rank dimension, higher r
increases expressivity (and memory/compute) but raises overfitting risk.  If you
raise r, consider stronger regularization or a lower learning rate.
* Scaling factor (α):  Scales the LoRA update; the effective update magnitude is
**alpha / r**, so setting alpha ≈ r keeps update strength stable.  Larger alpha
amplifies adaptation but can destabilize training if LR is high.
* Dropout:  Probability of dropping the adapter path to regularize training; higher
dropout curbs overfitting, especially with large r or small datasets.  With higher
dropout, you can often afford a slightly larger alpha or LR without instability.
* Batch size:  Number of samples per optimizer step|larger batches give smoother
gradients and typically permit a proportionally larger learning rate (linear-scaling
rule) at the cost of more memory.  Small batches may need gradient accumulation or a
reduced LR.
* Learning rate:  Step size for adapter parameters|too high can diverge (especially
with large alpha/r), too low slows convergence.  Tune in conjunction with batch size
and consider schedules (e.g., cosine) to balance speed and stability.
* rank(r):  {rank_value}
* Scaling factor(α):  {alpha_value}
* Dropout Rate:  {dropout_value}
* Batch Size:  {batchsize_value}
* Learning Rate:  {lr_value}
```

*(b)* Prompt templates with domain-aware prompting

