# OpenReview forum: "A Language-Guided Bayesian Optimization for Efficient LoRA Hyperparameter Search"
_ICML.cc/2026/Conference — ICML 2026 regular_

### Official Review · Reviewer_H213 · 2026-02-17

**Soundness:** 3
**Presentation:** 3
**Significance:** 3
**Originality:** 3
**Overall Recommendation:** 4
**Confidence:** 4

**Summary:**

This work proposed a hyperparameter optimization method for LoRA under Bayesian framework. A frozen pretrained LLM is used to map discrete hyperparameter configurations to a continuous embedding space in  which the Bayesian framework is applied. This paper uss domain-aware prompting, learnable token, and projection layer to calibrate the embedding verctors for hyperparamater encoding.

**Compliance With Llm Reviewing Policy:**

Affirmed.

**Key Questions For Authors:**

1) The proposed method can select good hyperparameters for LoRA. Can the authors provide insights on the optimally selected parameters?

2) The authors should conduct experiments on more tasks. Now the effectiveness is seen on GSM8K, MATH, HumanEval, MBPP. Only four are not enough

**Limitations:**

1) The proposed method can select good hyperparameters for LoRA. Can the authors provide insights on the optimally selected parameters?

2) The authors should conduct experiments on more tasks. Now the effectiveness is seen on GSM8K, MATH, HumanEval, MBPP. Only four are not enough

**Strengths And Weaknesses:**

***Strength***
1) The problem of LoRA's sensitivity to hyperparameters is important. Better hyperameter selection is important to the success of LoRA.

2) The experiments are conducted with not only original version of LoRA, but also with LoRA's variants including rsLoRA, DoRA, PiSSA. And the experiments are done on difference base models.

3) The experiment result is promising compared to the default hyperparameters configuration and existing hyperparameter searching method.

***Weakness***
1) The proposed method can select good hyperparameters for LoRA. Can the authors provide insights on the optimally selected parameters?

2) The authors should conduct experiments on more tasks. Now the effectiveness is seen on GSM8K, MATH, HumanEval, MBPP. Only four are not enough

---

> ### Author Rebuttal · Authors · 2026-03-31
>
> We thank the reviewer for the comments. We address the following two points: (W1) Insights into LoRA hyperparameters and (W2) Additional evaluation on MT-Bench.
>
> \
> **W1. Insights into LoRA hyperparameters**\
> We note that the main paper already discusses the hyperparameter trends identified by our method (page 5, line 267; page 8, line 409). For clarity, we briefly reiterate these findings here.
>
> > [Line 267] We further analyze the hyperparameter combinations discovered in our experiments. Smaller batch sizes are often preferred, consistent with prior findings (Marek et al., 2025). In addition, applying dropout often leads to better performance. Interestingly, we sometimes observe strong performance when the scaling factor (α) is 16 or even 32 times larger than the rank. This observation has not been reported in prior studies, where α was set to twice the rank according to existing guidelines (Diehl, 2024; unsloth, 2025), or determined based on a rank or fixed α (Kalajdzievski, 2023; Sun et al., 2024; Liu et al., 2025).
>
> > [Line 408] Compared to LLaMA2, Qwen2.5 models tend to prefer smaller ranks and larger batch sizes.
>
> Overall, these results suggest that effective LoRA hyperparameters cannot be explained by a single universal combination. Rather, the preferred configuration depends on the base model and the task. We believe this directly motivates the need for model- and task-specific hyperparameter search, which is precisely the goal of our method.
>
> \
> **W2. Additional evaluation on more tasks**\
> As requested, during the rebuttal period, we have added an additional evaluation of our method on MT-Bench. MT-Bench is a multi-turn conversational benchmark designed to assess instruction following, reasoning, and response quality in open-ended dialogue. This benchmark is qualitatively distinct from math reasoning and code generation. Following [C1], we use the standard MT-Bench protocol and report GPT-4 based GPT-scores.
>
> \
> **[Results across LoRA variants]**
> | Strategy | MT-Bench (Default) | MT-Bench (Ours) |
> |---|---:|---:|
> | LoRA | 7.181 | **7.350 (+0.169)** |
> | DoRA | 7.300 | **7.662 (+0.362)** |
> | rsLoRA | 7.125 | **7.475 (+0.350)** |
> | PiSSA | 7.200 | **7.475 (+0.275)** |
>
> **[Results across base models]**
> | Base Model | MT-Bench (Default) | MT-Bench (Ours) |
> |---|---:|---:|
> | LLaMA2-7B | 7.181 | **7.350 (+0.169)** |
> | Mistral-7B-v0.1 | 8.425 | **8.688 (+0.263)** |
> | Gemma-7B | 8.363 | **8.488 (+0.125)** |
>
> The results show that our method consistently outperforms the default hyperparameters reported in prior work [C1, C2] across all four adaptation strategies and all three base models. Since MT-Bench evaluates open-ended conversational ability, these results provide additional evidence that the effectiveness of our method is not limited to math reasoning and code generation, but also extends to other downstream tasks. We will include these results in the final version to broaden the empirical coverage of the paper.
>
> [C1] Meng, Fanxu, Zhaohui Wang, and Muhan Zhang. "Pissa: Principal singular values and singular vectors adaptation of large language models." NeurIPS, 2024.\
> [C2] Wang, Zhengbo, et al. "Lora-pro: Are low-rank adapters properly optimized?." ICLR, 2025.
>
> \
> We welcome any further comments. If our responses are satisfactory, please consider reassessing and upgrading the rating.

---

> > ### Author Rebuttal · Reviewer_H213 · 2026-04-01
> >
> > Thank you for your explanation. i have changed my score

---

> > > ### Author Response · Authors · 2026-04-06
> > >
> > > We sincerely thank you for reconsidering our work and raising your score! Your suggestion to conduct an additional benchmark significantly strengthened our empirical validation. We will ensure these results are integrated into the final version. If there are any further questions or comments, please feel free to let us know. Thank you again for your positive engagement.

---

### Official Review · Reviewer_LjJy · 2026-03-11

**Soundness:** 2
**Presentation:** 3
**Significance:** 2
**Originality:** 2
**Overall Recommendation:** 4
**Confidence:** 3

**Summary:**

The paper proposes an LLM-guided hyperparameter optimization method for LoRA using Bayesian Optimization with a Gaussian Process surrogate with a deep kernel. To embed the hyperparameter configurations, values of corresponding hyperparameters are inserted into a template to create a prompt. Then, a learnable token is appended to the prompt, passed through the LLM, and the final-token hidden state is extracted. Finally, the hidden state is projected to get the embedding for the GP.  The LoRA domain knowledge is reflected in the design of the search space and prompt template.

The GP, which is trained jointly with the projection layer and the token to predict performance. To speed up evaluation, the method uses a proxy benchmark where LoRA is trained on a subset of the dataset. The HPO and LoRA are performed on MetaMathQA and CodeFeedback, and evaluated on GSM8K / MATH and HumanEval / MBPP correspondingly. Each of the training sets contains 100K samples, with a 10K subset used for proxy training evaluation. The paper analyzes method and proxy design choices through ablations and compares the method with general and LoRA-specific HPO baselines across different model sizes and LoRA variants. The results show that the proposed method outperforms general and LoRA-specific HPO, while using less compute.

**Compliance With Llm Reviewing Policy:**

Affirmed.

**Final Justification:**

The authors have mostly addressed my concerns during the rebuttal. The method is interesting and can be published, which is why I increased my score to a weak accept. The paper could be more thorough by focusing more on the Bayesian optimization aspect, which is why I did not increase to a full accept.

**Key Questions For Authors:**

1. Please clarify the methodological contributions relative to prior work, especially with regard to [1].
2. The evaluation is comprehensive, but it seems to rely on the assumption of positive transfer between MetaMathQA and GSM8K/MATH, and between CodeFeedback and HumanEval/MBPP. Could the authors clarify whether negative transfer might affect the results, potentially making a worse-performing HPO algorithm appear favorable? Would it be possible to either perform transfer within each dataset of the group or at least additionally run HPO directly on the holdout datasets to confirm that the reported improvements are not artifacts of the evaluation setup?

**Limitations:**

I think the limitations regarding the results of the transfer between tasks, as well as the limited size of the evaluation (2 transfers from 2 datasets), should be discussed

**Strengths And Weaknesses:**

## Soundness:
The methodology is sound, and the evaluation is comprehensive; however, the evaluation is done on different datasets than the LORA is finetuned on. Thus,  it relies on the assumption of positive transfer between MetaMathQA and GSM8K / MATH, and between CodeFeedback and HumanEval / MBPP. If the transfer was negative, a worse-performing HPO algorithm could appear favorable under the proposed evaluation protocol. It would therefore be more appropriate to perform a transfer across each dataset within the groups, and additionally run HPO directly on the holdout datasets. This would help demonstrate that the selected tasks indeed exhibit positive transfer and that the reported improvements are not an artifact of the evaluation setup

## Presentation:
* The paper is clearly written and easy to follow.
* line 162 "The surrogate model parameterized ω is updated by maximizing"

## Significance:
The problem addressed in the paper is well motivated, and the proposed method, particularly the domain-aware prompt template, is interesting. However, the problem formulation itself appears somewhat narrow, focusing on hyperparameter optimization for LoRA. In practice, LoRA-specific design choices seem to play a role mainly in the prompt design used to encode hyperparameter configurations, which is also acknowledged in the Broader Impact section.
Consequently, the proposed approach appears more general than suggested: with an appropriately structured prompt that captures relevant domain knowledge, the method could likely be applied to other machine learning methods requiring hyperparameter optimization. Clarifying this broader applicability would strengthen the significance of the work.

## Originality:
* The contribution appears largely incremental, as most of the methodological components already exist and have previously been explored in the context of Bayesian Optimization:
    * The use of LLM embeddings for BO is conceptually very similar to work on prompt optimization, which should be mentioned [2].
    * The use of a learnable token is related to techniques from Soft Prompting Optimization.
    * The proposed proxy evaluation on a subset of data to accelerate HPO is also a common strategy, although usually multiple proxy levels are defined, and the best performing configuration is also evaluated on the highest fidelity level [see 3 for example methods, e.g., Successive Halving, Hyperband, or BOHB]. Therefore, while it appears to work for the given problem, it seems to be completely heuristic and should not be touted as a major contribution (at least not without a thorough comparison against prior work).
* Overall, the method closely resembles the PLLM variant of the approach proposed in GOLLuM [1]. In this sense, the main difference appears to be its application to the LoRA fine-tuning setting. The paper could highlight the generality of the proposed method in incorporating prior knowledge in the hyperparameter optimization process.
* It would therefore be helpful if the authors clarified the precise methodological contributions relative to this prior work.

[1] Ranković, Bojana, and Philippe Schwaller. "Gollum: Gaussian process optimized llms—reframing llm finetuning through bayesian optimization." ICLR 2025 Workshop on World Models: Understanding, Modelling and Scaling. 2025.

[2] Hyperband-based Bayesian Optimization for Black-box Prompt Selection. Lennart Schneider, Martin Wistuba, Aaron Klein, Jacek Golebiowski, Giovanni Zappella, Felice Antonio Merra Proceedings of the 42nd International Conference on Machine Learning, PMLR 267:53413-53438, 2025.

[3] Hyperparameter Optimization in Machine Learning. Luca Franceschi, Michele Donini, Valerio Perrone, Aaron Klein, Cédric Archambeau, Matthias Seeger, Massimiliano Pontil, Paolo Frasconi. arXiv:2410.22854 [stat.ML]

---

> ### Author Rebuttal · Authors · 2026-03-31
>
> We thank the reviewer for the constructive comments. We address the three key points.
>
> **Q1&Originality. Clarifying our contributions**
>
> We would like to emphasize that our contribution and novelty lie in integrating BO, LLMs, and other relevant components into a unified LoRA HPO framework. As noted by Reviewers GcHW and H213, since LoRA HPO is an important and timely problem, integrating them into an effective framework for LoRA HPO itself constitutes a meaningful contribution, while our framework builds on known components.
>
> We further clarify our methodological contributions in comparison to GOLLuM and prior work.
> - **Domain-aware prompting**: Unlike GOLLuM and Hyperband-based BO, which embed generic text, our method explicitly injects knowledge about the roles and relationships of LoRA hyperparameters. This gives the embedding space task-relevant structure, and the effectiveness of such a description has never been investigated before.
> - **Learnable token**: We propose the LoRA HPO-specific soft prompting mechanism integrated into Deep Kernel Learning (DKL)-based BO. It injects structural prior information into the embedding space. This is the first to show the compatibility and favorable performance of soft prompting in DKL-based BO.
> - **Proxy training**: This is the strategy to speed up the overall framework. Since our proposed framework shows better performance and effectiveness, the combination with proxy training offers a better balance that the existing research has not shown before.
>
> These technical contributions were also recognized by other reviewers. Reviewers GcHW and 9cPT highlighted the value of our domain-aware prompting and learnable tokens, noting that these components effectively smooth and structure the embedding space for efficient optimization. Reviewer 9cPT also noted that our proxy training strategy significantly improves efficiency while maintaining a high correlation with full-dataset performance.
>
> In summary, our method goes beyond a heuristic combination of known techniques and demonstrates that problem-specific integration can lead to both performance gains and practical efficiency over existing HPO approaches.
> We believe this integration is novel because our proposed method exhibits consistent effectiveness over any existing methods, which have not been achieved before. This fact highlights our clear and novel contribution.
>
> **Q2&Soundness. Validating the transfer assumption**\
> Our original experiment setup follows the prior work on LoRA variants [C1], which adopts the transfer assumption. To directly address the reviewer’s concern, we additionally conduct HPO on the holdout dataset (GSM8K).
> | Model | Default | Ours |
> |:---:|:---:|:---:|
> | LLaMA2-7B | 15.54 | 39.42 |
> | Qwen2.5-7B | 79.08 | 83.47 |
>
> For LLaMA2-7B, we use the default hyperparameters reported in our paper. For Qwen2.5-7B, since the default setting is not separately evaluated in the main text, we follow [C2]. In both models, our method outperforms the default setting even when optimization is conducted on the holdout dataset. These results show that the performance gains are not artifacts of a particular transfer setting, but rather stem from more effective HPO.
>
> **Significance. Generality beyond LoRA**\
> As the reviewer noted, our method is not limited to LoRA-specific settings. With a properly structured prompt that captures relevant domain knowledge, it can be applied to other setups as well. Nevertheless, as noted by Reviewers GcHW and H213, LoRA HPO is an important and timely problem that remains underexplored despite widespread LoRA adoption, which motivated us to focus on this task.
>
> To demonstrate the generality of our method and address the reviewer’s concern, we additionally evaluate our pipeline on Bayesmark [C3] using two classification datasets and three models.
>
> | Dataset | Model | Random | BO | Optuna | LLAMBO | Ours |
> |:---|:---|:---:|:---:|:---:|:---:|:---:|
> | **Breast Cancer** | Random Forest | 92.97 | 95.61 | 92.97 | 95.79 | 96.14 |
> | | MLP | 97.37 | 98.07 | 98.07 | 97.72 | 97.89 |
> | | Decision Tree | 91.38 | 93.85 | 91.74 | 91.74 | 94.38 |
> | **Iris** | Random Forest | 94.67 | 96.67 | 96.67 | 95.33 | 96.67 |
> | | MLP | 95.33 | 97.33 | 96.00 | 97.33 | 97.33 |
> | | Decision Tree | 93.33 | 96.00 | 94.00 | 94.00 | 96.00 |
>
> Overall, our method achieves the best or tied-for-best result in five of six settings, demonstrating the strongest overall performance among HPO methods. These results suggest that our framework generalizes beyond LoRA HPO. We will revise the paper to make this broader applicability clearer, while emphasizing that LoRA HPO is itself an important problem.
>
> [C1] Meng, et al. "Pissa: ..." NeurIPS, 2024.\
> [C2] Rathore, et al. "How Much is Too Much?..." AACL IJCNLP, 2025.\
> [C3] Uber, “bayesmark: ...,” 2020. Github repository.
>
> \
> We welcome any further feedback. If our responses are satisfactory, please consider reassessing and upgrading the rating.

---

> > ### Author Rebuttal · Reviewer_LjJy · 2026-04-01
> >
> > Thank you very much for providing further clarification. Most of the provided results make sense, and I agree that the domain of LORA is quite important. Your emphasis on the contribution of a method for LORA made me remember the paper AutoPEFT (https://aclanthology.org/2024.tacl-1.29.pdf) that should be discussed, as there have been other works on optimizing LORA hyperparameters before.
> >
> > Besides this, there are two things that I do not (yet) understand:
> > 1. what is the reason why you highlight the proxy training so much? This is a very common method in hyperparameter optimization, and I am actually somewhat worried that the proposed method only tunes hyperparameters on the proxy method and does not follow something more established such as the HyperBand schedule (and the combination of BayesOpt and HyperBand in BOHB). To the best of my knowledge, there is also no domain where adding more data decreases the performance, so the performance of the proxy is to be expected.
> > 2. Why would it be a good idea to show the capabilities of a hyperparameter optimization method by tuning an MLP for Iris? In my opinion, this does not present anything meaningful, as Iris is really a toy task.

---

> > > ### Author Response · Authors · 2026-04-06
> > >
> > > Thank you for recognizing the importance of the LoRA setting and follow-up questions. We address the reviewer’s questions below.
> > >
> > > \
> > > **1. Proxy training**
> > >
> > > As the reviewer noted, proxy training itself has been used in prior HPO literature. However, our emphasis is not on the novelty of the proxy training itself. Rather, we focus on analyzing how effective the integration of the proxy strategy and the particular setting of **LLM-guided BO framework for LoRA HPO** is and on understanding the behavior gap between the proxy training and the full training in practice, which has never been experimented with in this regime.
> > >
> > > We agree with the intuition that using more data generally does not reduce performance. However, the key point in proxy training is not the absolute performance itself. The more important point is whether the relative trends across hyperparameter configurations observed on the subset are preserved after full-data training. From this perspective, we empirically verify that only 10% subset provides a strong proxy signal, showing high correlation with full-dataset performance (Table 7 in the main text). This suggests that even a simple proxy can be sufficient and stable in this regime, without requiring more complex scheduling mechanisms.
> > >
> > > In conclusion, what we want to emphasize is not the proxy strategy itself, but the empirical validation of its usefulness in this setting, together with the effectiveness of combining it with our method. We will address this concern by revising the paper to tone down any wording that may present proxy training itself as an independent contribution, and more clearly emphasize the empirical validation and the benefits of integrating it into our method.
> > >
> > > \
> > > **2. Experiments on Bayesmark**
> > >
> > > We included the Iris-MLP setup because it is one of the well-known and **standard** dataset-model pairs in Bayesmark and has been widely used in previous HPO studies [C1-C4], which makes it a natural reference point for comparison. However, to address the reviewer’s concern more directly, we additionally evaluated our method on another Bayesmark task, ''Diabetes'', using three different models.
> > >
> > > | Dataset | Model | Random | BO | Optuna | LLAMBO | Ours |
> > > |:---:|:---|:---:|:---:|---|---|---|
> > > | Diabetes | Random Forest | -0.6468 | -0.5343 | -0.6010 | -0.5320 | **-0.5313** |
> > > |  | MLP | -0.4928 | -0.4877 | -0.4941 | -0.4965 | **-0.4859** |
> > > |  | Decision Tree | -0.6926 | -0.6547 | -0.7206 | -0.6696 | **-0.6240** |
> > >
> > > The experimental results show that our method achieves the best negative MSE score across all three models, and the same overall trend observed in the earlier experiments is preserved. This suggests that the effectiveness of our method is not specific to the Iris-MLP setup, but extends more consistently to other datasets and models.
> > >
> > > We will include the Diabetes results with the existing Breast Cancer and Iris results. In addition to LoRA results, we believe these will make the empirical evidence clearer and better demonstrate that our method is effective beyond the Iris-MLP setup.
> > >
> > > \
> > > [C1] Turner, Ryan, et al. "Bayesian optimization is superior to random search for machine learning hyperparameter tuning: Analysis of the black-box optimization challenge 2020." NeurIPS 2020 competition and demonstration track. PMLR, 2021.
> > >
> > > [C2] Cui, Lei, et al. "Neighbor Regularized Bayesian Optimization for Hyperparameter Optimization." BMVC, 2022.
> > >
> > > [C3] Cowen-Rivers, Alexander I., et al. "Hebo: Pushing the limits of sample-efficient hyper-parameter optimisation." JAIR, 2022.
> > >
> > > [C4] Liu, Tennison, et al. "Large language models to enhance bayesian optimization." ICLR, 2024.
> > >
> > > \
> > > We also thank the reviewer for pointing out AutoPEFT, which we will include in the related work section of the final version. If there are any further questions or comments, we would be happy to discuss them.

---

### Official Review · Reviewer_9cPT · 2026-03-11

**Soundness:** 4
**Presentation:** 4
**Significance:** 4
**Originality:** 3
**Overall Recommendation:** 5
**Confidence:** 2

**Summary:**

LLMs have strong potential and ability to serve as foundational models for downstream tasks, but the cost of fully fine-tuning LLMs to be capable of new tasks is computationally intensive. To tackle this problem, many parameter-efficient training techniques have emerged, and LoRA is one of the most widely used methods. Despite the success of LoRA, later research points out that it is highly sensitive to the selection of adapter hyperparameters. Different choices of hyperparameters can result in highly biased results. To obtain optimal results, finding an optimal hyperparameter combination for LoRA is important. An exhaustive search is infeasible due to the enormous search space, and naïve exploration is equally impractical. Therefore, the authors introduce Bayesian Optimization to address this issue. BO is effective at approximating the black-box function defined by the hyperparameters, making it well-suited for optimal hyperparameter searching in LoRA.
Although BO is powerful, it is not trivial to directly apply it to the LoRA hyperparameter optimization problem. This is because BO does not contain domain knowledge about the problem and requires the optimization function to be continuous. To address these limitations, the authors propose an efficient BO framework to the LoRA HPO problem. The proposed framework leverages the prior knowledge in LLMs to enhance the optimization process. For efficiency improvements, a proxy training evaluation process is introduced to reduce cost and time. Experimental results demonstrate consistent performance improvements, showing that the proposed framework is both more efficient and more effective than existing search methods.

**Compliance With Llm Reviewing Policy:**

Affirmed.

**Final Justification:**

The author's response addressed my concern.

**Key Questions For Authors:**

No

**Limitations:**

No limitations and future works are included in the discussion.

**Strengths And Weaknesses:**

Strengths:
1.BO relies on a continuous and smooth function space for efficient searching, whereas LoRA HPO operates in a discrete space. The authors successfully map the discrete hyperparameter set into a continuous vector space. This is achieved by feeding structured textual information into an LLM and obtaining embeddings by pooling the hidden state at the last token position, which are then passed through a projection layer to produce the final feature z. This feature z serves as the continuous and smooth search space for BO. To leverage rich domain knowledge, the input textual information follows a specific template that describes the relationships between hyperparameters and their respective roles.
2. A learnable token and a learnable projection layer are proposed through deep kernel learning. Figure 1 illustrates the effectiveness of introducing these learnable parameters. Together, they produce a smoother and more structured embedding space, which enables more efficient optimization.
3. Proxy training evaluation is introduced to improve efficiency. Using only 10% of randomly sampled training data, the search remains sufficient and achieves comparably high correlation with full-data results.

Weaknesses:
•The textual domain knowledge is manually specified, drawing from previous papers and empirical experience. I am wondering what would happen if we did not provide the LLM with this information and instead let it learn the relationships between hyperparameters on its own. I feel like it might be worth exploring an approach similar to the idea behind AlphaGo — only providing evaluation metrics as feedback, with no human prior knowledge guidance at all.

•Typos and other small mistakes:

In section 4.1 Experiment setting-LoRA hyperparameters and setup. In sentence: “We define the candidate pool of loRA hyperparameters as shown in Table 1.” It should be LoRA instead of loRA.

---

> ### Author Rebuttal · Authors · 2026-03-31
>
> We thank the reviewer for the insightful suggestion and for pointing out the typos. We address the reviewer's suggestion directly.
>
> \
> **Effect without textual domain knowledge extracted from previous research**\
> As requested, we have conducted additional experiments where no handcrafted domain-aware prompting is used.
> Instead, the LLM is guided by information learned directly from the optimization loop. Specifically, as suggested, we newly design an automatic alternative method in which the surrogate model is trained using only evaluation metrics to capture the relationship between hyperparameters and performance. These learned trends are then converted into textual descriptions by GPT, which replace the manually designed domain-aware prompting in subsequent steps. This process enables us to inject hyperparameter structure into the LLM without human intervention.
>
> For comparison, we evaluate four settings: (1) default hyperparameters, (2) our method without domain-aware prompting, (3) our main method with domain-aware prompting based on human prior knowledge, and (4) our new method with automatically generated domain-aware prompting.
>
> | No. | Ours | Domain-aware prompting | Manual | GSM8K | MATH |
> |:---:|:---:|:---:|:---:|:---:|:---:|
> |(1)| X | X | - | 47.76 | 8.72 |
> |(2)| O | X | - | 53.98 | 9.16 |
> |(3)| O | O | O | 61.41 | 12.46 |
> |(4)| O | O | X | 55.73 | 10.48 |
>
> \
> Interestingly, the results show that the automatically generated domain-aware prompting, Method (4), outperforms both the default hyperparameter setting and our method without domain-aware prompting. This indicates that useful hyperparameter structure can be learned automatically from evaluation metrics alone, without manually specified templates.
> However, our final design choice, which uses domain-aware prompting based on human prior knowledge, achieves the best performance, suggesting that the human-processed knowledge serves as a promising inductive bias.
>
> Overall, these results show that our framework remains effective even without human knowledge, while benefiting further from it when available. These also suggest that our method is practical in scenarios where domain knowledge is limited or unavailable.
>
> \
> We again thank the reviewer for this valuable suggestion that improves our work, and we will include this analysis in the final version to better clarify the role of domain knowledge.

---

> > ### Author Rebuttal · Reviewer_9cPT · 2026-04-03
> >
> > Thanks for the responses and I really appreciate the additional experiments. Good luck.

---

> > > ### Author Response · Authors · 2026-04-06
> > >
> > > Thank you for your thoughtful suggestion and positive engagement!
> > > Your insightful suggestion has significantly helped us improve the quality of our paper. If there are any further questions or additional comments, please feel free to share them.
> > > Thank you again for your support.

---

### Official Review · Reviewer_GcHW · 2026-03-13

**Soundness:** 3
**Presentation:** 2
**Significance:** 3
**Originality:** 3
**Overall Recommendation:** 4
**Confidence:** 3

**Summary:**

This paper proposes a language-guided Bayesian Optimization framework for LoRA hyperparameter search. The core idea is to use a frozen LLM to map discrete hyperparameter configurations into a continuous embedding space via domain-aware text prompting, then perform BO in that space using a Gaussian Process surrogate with deep kernel learning. A learnable token and a projection layer calibrate the embedding space for BO. The authors leverage a proxy training evaluation strategy (training on 10% of data) to further reduce cost. The framework is evaluated across multiple LoRA variants, model architectures, and tasks, showing consistent gains over baselines under the same exploration budget. The authors conduct extensive ablations to better underline the strengths of their method.

**Compliance With Llm Reviewing Policy:**

Affirmed.

**Final Justification:**

The rebuttal did strengthen the submission (e.g. W1 and the compute efficiency curve), but I opine that upgrading from Weak accept to Accept would require a clear mitigation of the remaining concerns and a proper integration of these new experiments to the paper that cannot reasonably be done within this cycle.

**Key Questions For Authors:**

- Could the surrogate model be leveraged to gather more general insights about hyperparameters in a "continuous" way? E.g. cases where a combination of hyperparameters is particularly strong or specific ranges / scaling laws for hyperparameters?

**Limitations:**

A discussion of limitations seems to be lacking. For instance, the reliance on an LLM and initial domain knowledge, or the complexity of the pipeline itself, could be acknowledged.

**Strengths And Weaknesses:**

Overall, this paper introduces an elaborate hyperparameter optimization method, justifies most of the components and shows their usefulness, and presents strong results in terms of performance and efficiency. Some additional experiments and reported results could improve the paper in terms of completeness and impact.

### Stengths

- **Timely and practically relevant**: Automated hyperparameter search for LoRA is an important and underexplored problem given how widely LoRA is deployed, and this is a well-timed contribution in the context of growing interest in automated research pipelines.
- **Strong empirical results**: The gains appear to be substantial and are consistent across LoRA variants (rsLoRA, DoRA, PiSSA), model architectures, and both math reasoning and code generation tasks.
- **Intriguing effectiveness of domain-aware prompting**: The ablation (Table 6) shows that domain-aware prompting is the single most impactful component, which is a genuinely interesting finding. It suggests that LLM latent representations can serve as a powerful prior for guiding HPO, an insight that opens up broader research directions.
- **Embedding space analysis**: The study presented in Figure 1 is insightful and it both strengthen the motivation for the specific implementation of the method and the understanding of the mechanism.

### Weaknesses
- **Missing efficiency curve**: The paper claims strong sample efficiency but does not provide a plot of performance versus compute/wall-clock time across iterations, integrating the (likely minimal) overhead of their prompting-based approach. Such a curve would make it much easier to assess at what point the framework saturates and how it compares to baselines at equal compute budgets.
- **No ablation on the choice of embedding LLM**: There is little discussion around the choice of the embedding model. First, it is unclear from the core of the paper what model is chosen, and the information is only disclosed in Appendix B, although it seems to me that this is a central implementation detail. Second, this choice raises several questions that remain unanswered, for instance, does a smaller LLM suffice? Does using the same LLM being fine-tuned (rather than a separate one) change the results? This is particularly relevant given the domain-aware prompting is so impactful.
- **Unclear nature of the embedding geometry**: It would strengthen the paper to characterize what the LLM embedding is actually capturing about hyperparameter configurations. Is the separation in Figure 1 driven by fine-grained feature filtering, or more coarse properties such as activation norms, a dominant PCA direction, or high-magnitude components? This analysis would clarify whether the LLM is doing something semantically meaningful or capturing high-level embedding statistics.

---

> ### Author Rebuttal · Authors · 2026-03-31
>
> We sincerely thank the reviewer for the constructive feedback. Below, we address the three weaknesses and one key question. All additional results and analyses below will be included in the final version.
>
> \
> **W1. Efficiency curve**\
> As requested, we newly measure cumulative wall-clock time and best-so-far performance at each BO iteration. The table below compares our method with the prompting-based approach (LLAMBO). To ensure a fair comparison, proxy training time is excluded.
>
> | Method | Iter | Time (s) | Subset (GSM8K / MATH, %) | Full (GSM8K / MATH, %) |
> |:---:|:---:|:---:|:---:|:---:|
> | **LLAMBO** | 5  | 31.7 | 26.7 / 16.9 | 55.0 / 9.2 |
> |  | 10 | 55.6 | 30.0 / 19.0 | 53.1 / 8.2 |
> |  | 15 | 82.2 | 30.0 / 19.3 | 54.1 / 8.4 |
> |  | 20 | 108.0 | 30.0 / 19.3 | 54.1 / 8.4 |
> |  | 25 | 133.4 | 30.0 / 19.3 | 54.1 / 8.4 |
> |  | 30 | 159.0 | 30.0 / 19.3 | 54.1 / 8.4 |
> | **Ours** | 5  | 11.7 | 33.7 / 4.3 | 50.4 / 7.2 |
> |  | 10 | 23.5 | 41.3 / 5.2 | 55.0 / 9.6 |
> |  | 15 | 35.9 | 44.2 / 6.2 | 56.4 / 11.4 |
> |  | 20 | 49.3 | 45.1 / 5.6 | 61.7 / 12.8 |
> |  | 25 | 61.6 | 45.0 / 6.1 | 62.9 / 12.9 |
> |  | 30 | 75.1 | 45.0 / 6.1 | 62.9 / 12.9 |
>
> Our method consistently achieves higher performance with lower cumulative time than LLAMBO. Moreover, while LLAMBO saturates early, our method continues improving across iterations, indicating more effective exploration. Overall, our framework improves both efficiency and effectiveness compared to the prompting-based baseline.
>
> \
> **W2. Ablation on the choice of embedding LLMs**
>
> We will clarify in the main text that we used Qwen2-7B as the primary embedding LLM.
>
> As requested, we further analyze the effect of the embedding model choice through two additional experiments: (1) varying the embedding model scale, and (2) using the same model as both the base model and the embedding model.
>
> | Base Model | Embedding Model | GSM8K (%) | MATH (%) |
> |:---:|:---:|:---:|:---:|
> | LLaMA2-7B | - | 41.47 | 5.24 |
> | LLaMA2-7B | Qwen2-1.5B | 54.21 | 9.82 |
> | LLaMA2-7B | Qwen2-7B | 62.93 | 12.88 |
> | LLaMA2-7B | LLaMA2-7B | 56.49 | 10.96 |
> | Qwen2.5-7B | Qwen2-7B | 83.93 | 48.08 |
> | Qwen2.5-7B | Qwen2.5-7B | 84.38 | 49.00
>
> First, even with a smaller embedding model (1.5B), our method outperforms the default hyperparameters, suggesting robustness to model scale.
> Second, using the same model for both roles still achieves better performance than using default hyperparameters, further demonstrating its effectiveness.
> These results demonstrate that our method does not depend on a specific embedding LLM while still benefiting from stronger, more advanced models.
>
> \
> **W3-1. Clarification of Figure 1**
>
> Sorry for the missing indication. Figure 1 shows a UMAP visualization of the learned latent representations for each hyperparameter configuration, extracted from LLM’s last hidden states. The overlaid color indicates the surrogate model’s predicted accuracy, since exhaustively evaluating all configurations is infeasible. The separation is not intentionally induced by feature filtering or coarse properties. Instead, it emerges naturally during LLM-based DKL training and appears in UMAP, consistent with the smoothness induced by the underlying Gaussian Process model in BO. We will make this explicit in the main text.
>
> \
> **W3-2&Q1. The analysis of embedding geometry**\
> As requested, to better characterize the embedding geometry, we’ve conducted an additional analysis to determine whether the surrogate model provides continuous, interpretable insights into promising hyperparameter regions.
>
> In the LLaMA2-7B setting, we applied PCA to the learned embeddings and sorted them by the first principal component. We then compared this ordering with surrogate-predicted performance. Among the top 300 configurations, we observed a strong monotonic relationship (Spearman's $rho$ = 0.8779), indicating that the dominant direction in the learned embedding space is well aligned with the objective and organizes configurations according to predicted performance. Furthermore, the analysis of the top 300 configurations along this axis reveals consistent patterns: larger rank (typically 64 or higher), smaller batch size (typically 8 or below), learning rates around $\mathrm{1e-4}$, and non-zero dropout are preferred. These trends align with the qualitative observations reported in line 267 of the main text.
>
> Overall, these findings suggest that the learned embeddings capture semantically meaningful structure in the hyperparameter space and that the surrogate model encodes both hyperparameter semantics and performance-relevant information. With more data, such analysis could become an even more powerful tool for understanding hyperparameter landscapes.
>
> In contrast, our analyses of activation norms and high-magnitude components did not find any such prominent signal. This is expected, given how embeddings are learned within the BO framework.
>
> \
> We again thank the reviewer for thoughtful feedback and welcome any further comments.

---

> > ### Author Rebuttal · Reviewer_GcHW · 2026-03-31
> >
> > Thank you for the response and the additional experiments. The rebuttal is valuable but the experiments do not extensively address my concerns and/or hint to changes that affect the overall narrative, and my score thus remains unchanged.I believe more effort would be necessary to make the additional experiments fully satisfactory:
> > - W1: more baselines would benefit these experiments. This should be a central result in the paper in my opinion.
> > - W2: this is an interesting result, but I do not agree with the statement: "These results demonstrate that our method does not depend on a specific embedding LLM". To my understanding, these results show that the final performance strongly depends on the chosen embedding model, even though all models lead to improvements over the baseline. This should be investigated further.
> > - W3.2 / q1: these experiments are interesting, but I believe that properly integrating them to the paper within this review cycle is not reasonable.

---

> > > ### Author Response · Authors · 2026-04-06
> > >
> > > We sincerely thank the reviewer for the feedback. Based on your comments, we conducted additional experiments and analyses. We address each point below.
> > >
> > > \
> > > **W1. Efficiency curve with more baselines**
> > >
> > > As requested, we have added all competing methods discussed in the paper to our efficiency experiments.
> > >
> > > | Method | Iter | Time (s) | Subset (GSM8K / MATH, %) | Full (GSM8K / MATH, %) |
> > > |:---:|:---:|:---:|:---:|:---:|
> > > | Ours | 5 | 11.7 | 33.7 / 4.3 | 50.4 / 7.2 |
> > > |  | 10 | 23.5 | 41.3 / 5.2 | 55.0 / 9.6 |
> > > |  | 15 | 35.9 | 44.2 / 6.2 | 56.4 / 11.4 |
> > > |  | 20 | 49.3 | 45.1 / 5.6 | 61.7 / 12.8 |
> > > |  | 25 | 61.6 | 45.0 / 6.1 | 62.9 / 12.9 |
> > > |  | 30 | 75.1 | 45.0 / 6.1 | 62.9 / 12.9 |
> > > | NOMAD | 5 | 18.6 | 39.3 / 4.7 | 53.1 / 8.8 |
> > > |  | 10 | 34.5 | 39.3 / 4.7 | 53.1 / 8.8 |
> > > |  | 15 | 54.7 | 41.2 / 5.1 | 54.4 / 9.7 |
> > > |  | 20 | 70.1 | 41.2 / 5.1 | 54.4 / 9.7 |
> > > |  | 25 | 85.7 | 41.2 / 5.1 | 54.4 / 9.7 |
> > > |  | 30 | 93.8 | 41.5 / 5.1 | 52.2 / 9.1 |
> > > | Latent BO | 5 | 8.1 | 32.0 / 4.4 | 57.7 / 11.7 |
> > > |  | 10 | 13.6 | 41.8 / 5.7 | 59.9 / 11.6 |
> > > |  | 15 | 20.5 | 45.0 / 6.1 | 59.5 / 11.9 |
> > > |  | 20 | 29.1 | 45.0 / 6.1 | 59.5 / 11.9 |
> > > |  | 25 | 40.0 | 45.0 / 6.1 | 59.5 / 11.9 |
> > > |  | 30 | 52.8 | 45.0 / 6.1 | 59.5 / 11.9 |
> > > | Naive BO | 5 | 2.6 | 31.3 / 3.5 | 50.8 / 7.8 |
> > > |  | 10 | 4.4 | 38.2 / 4.4 | 50.6 / 8.1 |
> > > |  | 15 | 6.2 | 41.2 / 5.1 | 57.8 / 10.8 |
> > > |  | 20 | 8.4 | 44.0 / 5.5 | 57.3 / 11.4 |
> > > |  | 25 | 10.8 | 44.0 / 5.5 | 57.3 / 11.4 |
> > > |  | 30 | 13.2 | 44.0 / 5.5 | 57.3 / 11.4 |
> > > | Optuna | 5 | 0.025 | 43.1 / 5.7 | 55.8 / 10.8 |
> > > |  | 10 | 0.045 | 43.1 / 5.7 | 55.8 / 10.8 |
> > > |  | 15 | 0.110 | 43.2 / 5.8 | 54.1 / 10.5 |
> > > |  | 20 | 0.146 | 43.2 / 5.8 | 54.1 / 10.5 |
> > > |  | 25 | 0.182 | 43.2 / 5.8 | 54.1 / 10.5 |
> > > |  | 30 | 0.219 | 43.2 / 5.8 | 54.1 / 10.5 |
> > > | Random | 5 | 0.064 | 37.8 / 4.6 | 52.8 / 8.4 |
> > > |  | 10 | 0.144 | 41.5 / 5.1 | 55.9 / 10.3 |
> > > |  | 15 | 0.217 | 41.5 / 5.1 | 55.9 / 10.3 |
> > > |  | 20 | 0.294 | 41.5 / 5.1 | 55.9 / 10.3 |
> > > |  | 25 | 0.369 | 41.5 / 5.1 | 55.9 / 10.3 |
> > > |  | 30 | 0.451 | 44.2 / 5.5 | 59.1 / 10.5 |
> > >
> > > The results show that our method consistently outperforms the Bayesian optimization-based approaches. We also observe that our method achieves both better performance and higher efficiency than NOMAD [C1], which was dedicatedly proposed for LoRA HPO. We believe these results reinforce that our contribution is not merely incremental but provides a meaningful improvement. We will revise the paper to highlight this as a central result.
> > >
> > > \
> > > **W2. The ablation of the embedding model**
> > >
> > > We appreciate comments regarding the embedding model. In addition to Qwen, we conducted further experiments using Mistral-7B-v0.2 and T5-base embedding models.
> > >
> > > | Base Model | Embedding Model | GSM8K | MATH |
> > > |:---:|:---:|:---:|:---:|
> > > | LLaMA2-7B | - | 41.47 | 5.24 |
> > > | LLaMA2-7B | Mistral-7B-v0.2 | 57.62 | 9.96 |
> > > | LLaMA2-7B | T5-base | 55.27 | 10.58 |
> > >
> > > The experimental results show consistent improvements over the default hyperparameter settings across all cases. While the final performance may vary across embedding models, which may require what embeddings are best or more favorable to our framework, we would like to emphasize that our framework can yield consistent gains regardless of the specific embedding model used. This suggests that the effectiveness of our method comes primarily from the pipeline itself, rather than from dependence on any particular embedding LLM.
> > >
> > > Beyond these results, we have also considered several hypotheses that may influence the effect of the embedding model, such as differences in hidden state dimensionality. A more systematic analysis of these hypotheses would require additional controlled experiments, which could not be comprehensively included within the rebuttal period. We will therefore include additional discussion on the role of embedding model choice in the final version.
> > >
> > > \
> > > **W3.2&Q1. The analysis of the embedding geometry**
> > >
> > > Thank you for suggesting this insightful analysis. We believe that this analysis offers a deeper understanding of the embedding space and complements our results. Therefore, we will include the key findings and insights in the appendix to provide additional context and interpretation.
> > >
> > > \
> > > [C1] Tribes, et al. "Hyperparameter optimization for large language model instruction-tuning." arXiv preprint arXiv:2312.00949 (2023).
> > >
> > > \
> > > We again thank the reviewer for the feedback and would be happy to address any further comments.

---

### Decision · Program_Chairs · 2026-04-30

**Decision:**

Accept (regular)

**Comment:**

This paper proposes an LLM-guided Bayesian optimization approach for hyperparameter search in LoRA. Reviewers found the aim of the paper to be timely and well-motivated due to the prevalence of LoRA. Additionally, experiments comprehensively demonstrate improved performance and efficiency across several benchmarks and LoRA variants. There were initial some concerns among the reviewers about whether the scope is overly narrow, as the method could potentially be applied beyond LoRA hyperparameter optimization, or that the novelty may have been slightly overstated. However, the authors addressed these concerns during the rebuttal phase. Due to the significance and soundness of the paper, I recommend acceptance.